


# Constraining Ammonia Emissions in Vehicle Plumes Utilizing Nitrogen Stable Isotopes

Wendell W. Walters[1,2], Linlin Song[3,4,5], Jiajue Chai[1,2], Yunting Fang[3,4,5], Nadia Colombi[1,*], and Meredith G. Hastings[1,2]

[1] Department of Earth, Environmental, and Planetary Sciences, Brown University, Providence, RI 02912, US
[2] Institute at Brown for Environment and Society, Brown University, Providence, RI 02912, US
[3] CAS Key Laboratory of Forest Ecology and Management, Institute of Applied Ecology, Chinese Academy of Sciences, Shenyang, Liaoning, 110016, China.
[4] Key Laboratory of Stable Isotope Techniques and Applications, Shenyang, Liaoning, 110016, China
[5] College of Sources and Environment, University of Chinese Academy of Sciences, Beijing 100049, China

*Current Address: Department of Earth and Planetary Science, Harvard University, Cambridge, MA 02139, US

*Correspondence to:* Wendell W. Walters (wendell_walters@brown.edu)

**Abstract.** Vehicle emissions have been identified as an important urban source of ammonia ($NH_3$). However, there are large uncertainties regarding the contribution of vehicle emissions to urban $NH_3$ budgets, as well as its role in spatiotemporal fine particulate matter ($PM_{2.5}$) formation and nitrogen (N) deposition patterns. The N stable isotopic composition ($\delta^{15}N$) may be a useful observational constraint to track $NH_3$ emission sources and chemical processing, but previously reported vehicle $\delta^{15}N(NH_3)$ emission signatures have reported a wide range of values, indicating the need for further refinement. Here we have characterized $\delta^{15}N(NH_3)$ spatiotemporal variabilities from vehicle plumes in stationary and on-road measurements in the US and China using a laboratory- and field-verified $NH_3$ collection technique shown to be accurate for characterizing $\delta^{15}N(NH_3)$ on the order of hourly time resolution. Significant spatial and temporal $\delta^{15}N(NH_3)$ variabilities were observed and suggested to be driven by vehicle fleet composition and influences from $NH_3$ dry deposition on tunnel surfaces. The reactive $NH_3$ sink associated with particulate ammonium ($pNH_4^+$) formation was found to have a minimal impact on the vehicle plume $\delta^{15}N(NH_3)$ measurements due to the vast majority of $NH_x$ ($= NH_3 + pNH_4^+$) residing as $NH_3$. Overall, a consistent $\delta^{15}N(NH_3)$ signature of 6.6±2.1 ‰ ($\bar{x}±1\sigma$; n=80) was found in vehicle plumes with fleet compositions typical of urban regions. Overall, these measurements constrain the $\delta^{15}N(NH_3)$ urban traffic plume signature, which has important implications for tracking vehicle $NH_3$ in urban-affected areas.

## 1. Introduction

Atmospheric ammonia ($NH_3$) is a critical component of the atmosphere (Behera et al., 2013) and the global nitrogen (N) cycle (Galloway et al., 2004). As the primary atmospheric alkaline molecule, $NH_3$ plays an essential role in the neutralization of sulfuric acid ($H_2SO_4$) and nitric acid ($HNO_3$), leading to the formation of ammonium nitrate ($NH_4NO_3$),



ammonium bisulfate ($NH_4HSO_4$), and ammonium sulfate (($NH_4)_2SO_4$) (Behera and Sharma, 2012).   These compounds are

the most abundant secondary components of inorganic fine particulate matter ($PM_{2.5}$), which has important implications for

air quality, human health, visibility, and global climate change (Behera and Sharma, 2010; Updyke et al., 2012; Wang et al.,

2015).   Deposition of $NH_3$ and its secondary product, particulate ammonium ($pNH_4^+$), have critical environmental

consequences, including acidification, eutrophication, and decreased biodiversity in sensitive ecosystems (Erisman et al.,

2008; Galloway et al., 2004; Sutton et al., 2008).   In recent years, N deposition in the form of $NH_x$ ($= NH_3 + pNH_4^+$) has

come to dominate total inorganic reactive N deposition across most of the United States (Li et al., 2016).   To evaluate the

influence of $NH_3$ on climate and the environment, an accurate understanding of $NH_3$ atmospheric concentrations, emission

sources, and spatiotemporal distributions are critical.   However, the quantification of $NH_3$ emission budgets remains

uncertain (Clarisse et al., 2009), and recent high-resolution satellite $NH_3$ observations imply that anthropogenic emission

inventories are substantially underestimated (Van Damme et al., 2018).

While agricultural activities are known to dominate the emission of $NH_3$, accounting for over 60 % of the global inventory

(Bouwman et al., 1997), there are significant spatiotemporal variabilities due to its short atmospheric lifetime and multitude

of emission sources (e.g., Hu et al., 2014).   In urban regions, vehicle derived emissions have been identified as a major $NH_3$

source (Gong et al., 2011; Li et al., 2006; Livingston et al., 2009; Meng et al., 2011; Nowak et al., 2012; Sun et al., 2014,

2017).   These emissions are mostly derived from light and medium-duty vehicles equipped with a three-way catalytic

converter (TWCC) (Bishop and Stedman, 2015), in which $NH_3$ is produced via steam reforming from hydrocarbons

(Whittington et al., 1995) and/or reaction of NO with molecular hydrogen (Barbier Jr and Duprez, 1994).   The recent

introduction of more stringent $NO_x$ reduction legislation has led to the incorporation of selective catalytic reduction (SCR)

systems in heavy-duty and light-duty diesel vehicles (Johnson, 2011).   This technology aims to reduce $NO_x$ emissions by

reacting it with $NH_3$, which is a product of the injected urea from diesel exhaust fluid (Bishop and Stedman, 2015;

Thiruvengadam et al., 2016; Gabrielsson, 2004).   Any overdosing of urea and/or catalytic degradation may lead to emissions

via "$NH_3$ slip", which has been detected in SCR equipped diesel vehicles in recent chassis dynamometer tests and on-road



measurements (Suarez-Bertoa et al., 2014; Suarez-Bertoa and Astorga, 2018; Suarez-Bertoa et al., 2017), indicating that the potential number of vehicles that could emit $NH_3$ may increase in the near future (Suarez-Bertoa et al., 2015).


While vehicle emissions may be a minor source of $NH_3$ on a global scale, these emissions can have a disproportionate impact on air quality and human health in urban regions due to high population density and the co-location of many $PM_{2.5}$ precursors including $NO_x$ and sulfur dioxide ($SO_2$) (Parrish and Zhu, 2009). Additionally, $NH_3$ is a crucial driver of N deposition, indicating the potential for traffic $NH_3$ emissions to impact N deposition in urban and urban-affected regions

(Fenn et al., 2018). However, identifying the source of increased urban $NH_3$ emissions, $pNH_4^+$ formation, and N deposition can be challenging due to the variety of potential emission sources that exist in the urban atmosphere including fuel combustion, waste containers, sewerage systems, transport from agricultural areas, and vehicles (Gong et al., 2011; Hu et al., 2014; Meng et al., 2011; Saylor et al., 2010; Sun et al., 2014, 2017; Sutton et al., 2000; Whitehead et al., 2007). The N stable isotopic composition ($\delta^{15}N$) of $NH_3$ could be a valuable observational constraint to track source contributions and

validate model apportionments (Felix et al., 2013; Felix et al., 2017). However, to quantitatively utilize this tracer requires further $\delta^{15}N(NH_3)$ source emission improvements and an increased understanding of spatiotemporal variabilities.

Tracking the contribution of vehicle $NH_3$ emissions might be possible using $\delta^{15}N(NH_3)$; however, previous measurements of vehicle $\delta^{15}N(NH_3)$ signatures are limited and have reported a wide range of values from -17.8 to 0.4 ‰ (Chang et al., 2016;

Felix et al., 2013; Smirnoff et al., 2012), which slightly overlaps with agricultural derived $NH_3$ that has been measured to range from -15.2 to -8.9 ‰ in animal-sheds (Heaton, 1987; Freyer, 1978). To quantitatively utilize $\delta^{15}N(NH_3)$ for $NH_3$ source apportionment requires distinguishable isotopic signatures, such that we need to understand the drivers behind the variability and determine whether it reflects actual source emission $\delta^{15}N(NH_3)$ inconsistencies or caused by other factors. The previous vehicle $\delta^{15}N(NH_3)$ characterization studies have included tunnel monitoring in the United States (Felix et al.,

2013), tunnel monitoring in China (Chang et al., 2016), and near-highway monitoring in Canada (Smirnoff et al., 2012), with reported $\delta^{15}N(NH_3)$ averages ($\bar{x}\pm1\sigma$) of -3.4±1.2 ‰ (n=2), -14.2±2.6 ‰ (n=8), and -2.1±1.9 ‰ (n=11), respectively. We



note that the observed variability may be related to spatiotemporal differences in the vehicle emitted $\delta^{15}N(NH_3)$, as the studies conducted in the US and Canada have reported relatively consistent values that are higher than reported for the study conducted in China, but the factors influencing this potential spatiotemporal $\delta^{15}N(NH_3)$ pattern are unknown (Chang et al.,

2016; Felix et al., 2013; Smirnoff et al., 2012). Notably, the reported $\delta^{15}N(NH_3)$ source measurements were conducted using a variety of $NH_3$ capture techniques for off-line $\delta^{15}N(NH_3)$ quantification that have included both passive samplers (Chang et al., 2016; Felix et al., 2013) and active collection using a filter pack (Smirnoff et al., 2012) for vehicle emission $\delta^{15}N(NH_3)$ characterization. Indeed, it has been shown that different active and passive $NH_3$ collection devices, including a gas-scrubbing bubbler, moss bag, shuttle sampler, and diffusion tube, resulted in significant $\delta^{15}N(NH_3)$ differences and variance

when sampling the same emission source (Skinner et al., 2006). None of these methods have been laboratory verified for their suitability for $\delta^{15}N(NH_3)$ analysis, and their accuracy remains unknown. Thus, there could be inaccuracies in the previously reported $\delta^{15}N(NH_3)$ emission values related to the collection technique used to concentrate ambient $NH_3$ for off-line $\delta^{15}N(NH_3)$ characterization.

Careful consideration of $NH_3$ sink processes associated with $pNH_4^+$ formation and $NH_3$ dry deposition must be taken into account, particularly under ambient environment sampling conditions, as these sink processes may alter the emission source $\delta^{15}N(NH_3)$ value (e.g., Felix et al., 2014; Skinner et al., 2006). However, the impact of $NH_3$ sink processes on previously reported vehicle $\delta^{15}N(NH_3)$ values are unknown (Chang et al., 2016; Felix et al., 2013; Smirnoff et al., 2012). Previous $\delta^{15}N(NH_3)$ characterization studies from agricultural transects in a dairy barn and intensive animal units have reported a

significant change in $\delta^{15}N(NH_3)$ within 50 m downwind of the emission sources, and this has been partly suggested to be driven by $\delta^{15}N$ fractionation during the conversion of $NH_3$ to $pNH_4^+$ (Felix et al., 2014; Skinner et al., 2006). Inevitably, downwind measurements conducted nearby $NH_3$ source emissions (i.e., within a few meters) would reduce the potential for $pNH_4^+$ formation and $NH_3$ dry deposition to have a strong influence on altering $\delta^{15}N(NH_3)$ values. For example, previous work has estimated a cumulative $NH_x$ deposition loss of less than 1% for a distance of 10 m downwind of $NH_3$ emissions

(Asman et al., 1998). However, the co-location of elevated $NH_3$ from vehicle emissions with available acidic components, including $H_2SO_4$ and $HNO_3$, may facilitate rapid $pNH_4^+$ formation (Behera and Sharma, 2011), which could have an





important influence on traffic plume $\delta^{15}N(NH_3)$ measurements. Additionally, samples collected in a tunnel could be influenced by enhanced $NH_3$ loss via dry deposition on tunnel surfaces (Sun et al., 2017), which might induce significant $\delta^{15}N(NH_3)$ fractionation. Thus, it remains unclear if tunnel $\delta^{15}N(NH_3)$ measurements are representative of all vehicle plume

$\delta^{15}N(NH_3)$ signatures, particularly well-ventilated environments such as open-highways.

$NH_3$ sink processes fractionate $\delta^{15}N(NH_3)$ values due to equilibrium and/or kinetic isotope effects. N isotopic equilibrium reactions between $NH_3$ and $NH_4^+$ will scramble the $^{14}N$ and $^{15}N$ isotopes, as demonstrated from theoretical estimates, laboratory experiments, and field observations (Kawashima and Ono, 2019; Savard et al., 2017; Urey, 1947; Walters et al.,

2018) (R1):

$$^{15}NH_{3\,(g)} + {}^{14}NH_{4\,(aq\,or\,s)}^+ \rightleftharpoons {}^{14}NH_{3(g)} + {}^{15}NH_{4\,(aq\,or\,s)}^+ \tag{R1}$$

The theoretical equilibrium constant (K) or isotopic fractionation factor (α) for R1 is near 1.035 at 26 °C (Urey, 1947; Walters et al., 2018).

$$\alpha\,(26\,°C) = \frac{{}^{15}NH_4^+/{}^{14}NH_4^+}{{}^{15}NH_3/{}^{14}NH_3} = 1.035 \tag{1}$$

This is equivalent to an isotopic enrichment factor (ε(‰)=1000(α-1)) of 35 ‰ indicating that isotopic equilibrium will favor the partitioning of $^{15}N$ into $pNH_4^+$ resulting in $\delta^{15}N(NH_3)$ values 35 ‰ lower relative to $\delta^{15}N(pNH_4^+)$. Additionally, a kinetic isotope effect (here defined as the ratio of the heavy to light isotope) has been suggested to occur during the initial incorporation of $NH_3$ into an aerosol as $pNH_4^+$, with an estimated enrichment factor of -28 ‰ based on the relative diffusion rates of the N isotopologues of $NH_3$ (Pan et al., 2016). This kinetic isotope effect would result in $\delta^{15}N(NH_3)$, which is 28 ‰

higher than the product $\delta^{15}N(pNH_4^+)$. The ability for $NH_3$ sink processes to influence source $\delta^{15}N(NH_3)$ signatures depends on the availability of acidic species (e.g., $HNO_3$, $H_2SO_4$), the amount of $NH_3$ dry deposited, and the driving isotopic fractionation process (i.e., equilibriums versus kinetic). Speciated measurements of $NH_x$, in different environments, could help quantify the influence of $NH_3$ sinks on altering $\delta^{15}N(NH_3)$ traffic source signatures and increase our understanding of tracking $NH_3$ emission sources from receptor monitoring locations utilizing $\delta^{15}N(NH_3)$.




Thus, to improve the $\delta^{15}N(NH_3)$ source inventory for accurate $NH_3$ source apportionment, we need to quantify $\delta^{15}N(NH_3)$ using accurate methods, identify the influence of $NH_3$ sink processes on $\delta^{15}N(NH_3)$, and address spatiotemporal variabilities. In this study, $\delta^{15}N(NH_3)$ was characterized from a variety of integrated vehicle plumes with a combination of stationary and mobile on-road measurements, utilizing a laboratory- and field-verified active collection technique shown to be accurate for

$\delta^{15}N(NH_3)$ quantification while providing $NH_x$ speciation (Walters and Hastings, 2018; Walters et al., 2019). Stationary measurements were conducted during the summer and winter at a near-highway monitoring site in Providence, RI, US, and within a tunnel in Shenyang, Liaoning, China. A broad spatial survey of on-road mobile measurements was also conducted in the northeastern US to evaluate the influences of a variety of real-world vehicle fleet compositions and driving modes on the traffic $\delta^{15}N(NH_3)$ signature. Passive $NH_3$ samplers, which have been used in previous $\delta^{15}N(NH_3)$ source characterization

studies (Chang et al., 2016; Felix et al., 2013, 2017), were also deployed in the near-highway and tunnel monitoring campaigns and compared with the active collection technique verified for $\delta^{15}N(NH_3)$ accuracy (Walters and Hastings, 2018). Overall, this data will better define the $\delta^{15}N(NH_3)$ source signature for urban vehicle plumes, with implications for tracking emission contributions to urban atmospheric $NH_3$ concentrations and N deposition.

## 2. Site Description and Methods

### 2.1 Sampling Sites

### 2.1.1 Near-Highway Measurements (Providence, RI, USA)

Stationary measurements were conducted at an air monitoring station in Providence, RI, US (41°49'46.0"N 71°25'03.0"W) maintained by the Rhode Island Department of Environmental Management (RI-DEM) and Rhode Island Department of Health (RI-DOH) (Figure S1) during the summer and winter. The air monitoring station is located 4.62 m east of

northbound I-95, a major interstate highway with a traffic volume of ~200,000 vehicles/day (HERE Traffic Analysis; https://company.here.com/automotive/traffic/traffic-analytics/), dominated by light-duty gasoline-powered vehicles. Collections of speciated $NH_x$ were conducted using a denuder-filter pack sampling device (ChemComb Speciation Cartridge; described in 2.2) with 6 h sampling intervals that included 00:30-6:30; 6:30-12:30; 12:30-18:30, and 18:30-00:30 during summer (August 9 to August 18, 2017) and 00:00-6:00, 6:00-12:00, 12:00-18:00, and 18:00-0:00 during winter (January 21



to February 1, 2018). During the sampling periods, $NH_x$ collections were not conducted during precipitation periods (or

forecasted precipitation periods) due to the potential role of wet scavenging to alter $\delta^{15}N(NH_3)$ (Xiao et al., 2015). The

sampling cartridges were mounted on the roof (~3.85 m above ground) of the air monitoring station on the underside of a

weatherproof shelter. The sampler's PTFE coated inlet was directly exposed to ambient air without the use of an additional

inlet tubing to prevent the loss of $NH_3$. Two passive samplers (ALPHA; Centre for Ecology & Hydrology) were deployed

during the winter for $NH_3$ collection to compare concentration and isotope results between passive and active collection

techniques. The ALPHA samplers were mounted on the underside of a weatherproof shelter and were exposed to ambient

air for two separate approximate 1-week collection periods during winter (Feb 10 – Feb 17 & Feb 17 – Feb 25 in 2018) for a

total of four collected samples at the near-highway monitoring site.

Ancillary on-line measurements of CO (Thermo Scientific 48i) were continuously monitored at the sampling location. This

instrument was housed in a climate-controlled trailer, and a sampling inlet was secured to the roof and weather protected.

Meteorological parameters, including temperature, relative humidity, wind speed, and wind direction, were recorded at the

Urban League RI-DEM monitoring site, 2.4 km south of the near-highway site (Figure S1). The wind measurements at

Urban League were assumed to be regionally representative of Providence, RI, US, as they are consistent with synchronous

hourly wind data from a meteorological station located at Brown University Ladd Observatory, located 4.0 km northeast of

Urban League (Miller et al., 2017) (Figure S1).

### 2.1.2 Tunnel Measurements (Shenyang, Liaoning, China)

From October 30 to November 5 in 2018, stationary tunnel measurements were conducted in the middle of an underground

tunnel of North-South Expressway in Shenyang, Liaoning Province, China (41°48'16.0"N 123°26'54.0"E). This tunnel is

approximately 2,360 m long, experiences approximately 28,804 vehicles/day during the weekday and 26,237 vehicles/day

during the weekend (data from real-time traffic control system, Shenyang WuAi Tunnel Management co. LTD). The tunnel

was open to vehicle passage from 5:00 to 23:00, and collections of speciated $NH_x$ were conducted using a denuder-filter pack





sampling device (ChemComb Speciation Cartridge) at 8 h intervals (approximately 6:00 to 14:00, 14:00 to 22:00, and 22:00

to 6:00). The sampling conducted from 22:00 to 6:00 included the period that the tunnel was closed to vehicle passage (i.e., 23:00 to 5:00). The denuder-filter pack samplers were mounted on an elevated platform approximately 1.5 m above ground and directly sampled ambient air within the tunnel (Figure S2). The platform was placed in the middle of the tunnel in a blocked off emergency lane. Three ALPHA samplers were also mounted on the elevated platform and simultaneously collected $NH_3$ during the sampling campaign (~7 days). The relative humidity and temperature within the tunnel were

monitored (iButton®, DS1923, Wdsen Electronic Technology Co., Ltd) from October 31, 2018, at 14:00 to the end of the sampling campaign that included measurements for 16 out of the 21 collection periods.

### 2.1.3 Mobile On-road Measurements in Northeastern US

Mobile on-road measurements were conducted in the northeastern US from February 20 to February 24, 2018, for

approximately 21 hours and spanned ~2,125 km. The mobile laboratory consisted of a pick-up truck (Ford F-150) equipped with a denuder-filter sampling device (ChemComb Speciation Cartridge), a CO analyzer (American Ecotech Serinus 30), a temperature and relative humidity probe (Elitech GSP-6), and a GPS tracking application (Map Plus) that recorded geolocation and vehicle speed (Figure S3). The denuder-filter pack samplers were placed in a weather-proof enclosure that was secured in the truck bed (~1 m above the truck bed) with the denuder sampling inlet directly exposed to ambient air to

limit the potential for $NH_3$ inlet loss, and collections were conducted for approximately 1 h. Sampling was temporarily ceased during periods in which our vehicle speed was lower than 15 km hr$^{-1}$ to limit the possibility of sampling self-emissions. The CO analyzer was placed inside the truck and kept at a similar temperature to calibration conditions in the laboratory, and an air sampling inlet (PTFE tubing, 6.35 mm OD) was secured to the roof of the truck (~1.9 m above ground). Due to the significant power demands (~600 W) of the on-board instruments and collection equipment (i.e.,

vacuum pump), a gasoline-powered generator (Champion 1200-Watt Portable Generator) was used to power all equipment. The exhaust from the generator was diverted and emitted alongside the truck exhaust.





## 2.2 Active collection of $NH_x$ using a denuder-filter pack

Active speciated $NH_x$ collection was conducted using a glass honeycomb denuder-filter pack sampling system (ChemComb
Speciation Cartridge) during all campaigns. This collection system has been extensively described for its ability to speciate
between reactive inorganic gases and particulate matter for off-line concentration determination (Koutrakis et al., 1988,
1993). Recently, this sampling system has been shown to enable speciated $\delta^{15}N(NH_x)$ quantification with a $\delta^{15}N(NH_3)$ and
$\delta^{15}N(pNH_4^+)$ precision ($\pm 1\sigma$) of $\pm 0.8$ ‰, and $\pm 0.9$ ‰, respectively (Walters et al., 2019; Walters and Hastings, 2018). The
sampler consisted of (-in order-) a PTFE coated inlet to minimize reactive gas loss, a $PM_{2.5}$ impactor plate, a basic-coated
honeycomb denuder (2% carbonate (w/v) + 1% glycerol (w/v) in 80:20 water:methanol (v/v) solution), acid-coated
honeycomb denuder (2 % citric acid (w/v) + 1 % glycerol (w/v) in 20:80 water:methanol (v/v) solution) to collect $NH_3$, and a
filter pack to collect $pNH_4^+$. The basic-coated denuder was used to remove atmospheric acids (e.g., $HNO_3$, $SO_2$, and
hydrochloric acid (HCl)) as a precaution to reduce collection-related gas-particle interactions. The gases collected on the
basic-coated denuder were generally below detection limits and were not reported in this work.


In the first measurement campaign (near-highway monitoring in summer of 2017), $pNH_4^+$ was collected using a single
Fluoropore PTFE membrane filter (Millipore, 1.0 µm pore, 47 mm diameter). However, due to potential loss of semi-
volatile $NH_4NO_3$, all subsequent campaigns utilized a Nylon filter (Cole-Parmer, 0.8 µm pore, 47 mm diameter) which has
been shown to collect and retain $pNO_3^-$ quantitatively (Yu et al., 2005), and a backup acid-coated (5 % citric acid (w/v) in
water) cellulose filter (Whatman, 8 µm pore, 47 mm diameter) to capture any volatilized $NH_3$ from the collected particles
and/or $NH_3$ breakthrough during conditions of denuder saturation (Walters et al., 2019; Yu et al., 2006). All collections were
conducted at a flow rate of 10 liters per minute (LPM) using a mass-flow controller (Dakota mass flow controller
6AGC1AL55-10AB2; precision $\pm 1\%$) attached to an oil-less vacuum pump (Welch 2546B-01). $NH_3$ was also collected
using a passive sampler (ALPHA), in which $NH_3$ diffuses through a PTFE membrane and accumulates on an acid-coated (5
% citric acid (w/v) in water) cellulose filter (Albet, Grade 604, 24 mm diameter) housed in a protective case.



All denuders and filters were cleaned and coated fresh daily. Denuder and filter coating solutions were produced using ACS certified reagents, and all water used for cleaning, coating, and extraction was of ultra-high purity grade (>18.2 MΩ). Denuders were coated using 10 mL of the appropriate solution and air-dried for at least 1 h. PTFE and Nylon filters were

rinsed and sonicated with 20 mL of water for 30 minutes three times and dried in an oven at 50 ℃ for at least thirty minutes. The cellulose filters were not rinsed before use due to disintegration issues and were directly impregnated with a 0.5 mL of the 5 % citric acid (w/v) solution and dried with ultra-high purity nitrogen (>99.9 %). The denuder-filter packs were loaded, and the inlet and outlet were capped and transported to the field site. Immediately after collection, the cartridges were capped and transported back to the laboratory for extraction. Periodically, field blanks were taken for all sample types (i.e.,

filters and denuders), representing approximately 10 % of the total number of collected samples. Replicate samples (e.g., side-by-side samples) were periodically conducted at the near-highway site during summer to determine measurement precision. The ALPHA samplers were prepared in the laboratory, capped, and then transported to the field site. After collection, the passive samplers were immediately capped and transported back to the laboratory for extraction.

Collections on the denuders and the acid-coated filters were extracted using water. The PTFE and Nylon filters were wetted with 500 μL of ethanol and then extracted in water. All filter samples were sonicated for 1 h. Filters were then removed using cleaned forceps, and the extraction solutions were then passed through a 0.22 μm syringe filter to remove loose filter pieces and potential microbial contaminants. After extraction, all solutions were placed in a freezer at -30 ℃ until subsequent concentration and isotopic analysis.


### 2.3 Concentration and $\delta^{15}N(NH_x)$ Isotopic Analysis

The concentrations of the denuder and filter extraction solutions were analyzed using a combination of colorimetry and/or ion chromatography analytical techniques. Colorimetric analysis included measurement of $[NH_4^+]$ based on the indophenol blue method with absorbance detection at 625 nm (e.g., US EPA Method 350.1), as well as $[NO_2^-]$ via diazotization with

sulfanilamide dihydrochloride followed by detection of absorbance at 520 nm (e.g., US EPA Method 353.2) that was





automated using a discrete UV-Vis spectrophotometer (Westco SmartChem Discrete Analyzer) at Brown University. These analyses were conducted for all samples collected in the US (i.e., near-highway and mobile measurements). Pooled standard deviations ($\pm 1\sigma$) of replicate measurements of quality control standards were $\pm 0.35$ µmol·L$^{-1}$ (n=48), and $\pm 0.23$ µmol·L$^{-1}$ (n=60), and the average relative standard deviations (RSD) were 1.3 % and 0.81 % for [NH$_4^+$] and [NO$_2^-$], respectively.   All

samples collected in the Shenyang tunnel were analyzed for [NH$_4^+$], [NO$_2^-$], [NO$_3^-$], and [SO$_4^{2-}$] using ion chromatography (Dionex™ ICS-600), at the Institute of Applied Ecology, Chinese Academy of Sciences. Cations were quantified using a Dionex™ CS12A column and CQ12A guard column with 10 mmol·L$^{-1}$ methanesulfonic acid as the eluent. Anions were quantified using a Dionex™ AS22 column and AQ22 guard column with 4.5 mmol·L$^{-1}$ sodium carbonate and 1.4 mmol·L$^{-1}$ sodium bicarbonate as the eluent. For all quantified ions, the RSD was less than 1.5 %. The measured [NH$_4^+$] was used to

calculate the concentrations of NH$_3$ and pNH$_4^+$ in the traffic plumes, while [NO$_2^-$] was quantified because it will interfere with nitrogen isotopic analysis of NH$_4^+$ (Zhang et al., 2007), but [NO$_2^-$] was never measured above the detection limit. Additional measurements of [NO$_3^-$] and [SO$_4^{2-}$] conducted for the Shenyang filter extracts were utilized as ancillary data.

The quantification of $\delta^{15}$N(NH$_4^+$) was performed separately for the acid-coated honeycomb denuder, the particulate filter,

and the acid-coated cellulose filter extraction solutions, corresponding to NH$_3$, pNH$_4^+$, and volatilized pNH$_4^+$ (and/or NH$_3$ breakthrough during denuder saturation conditions), respectively. Briefly, $\delta^{15}$N(NH$_4^+$) was measured based on an established off-line wet-chemistry technique involving hypobromite (BrO$^-$) oxidation and acetic acid/sodium azide reduction (Zhang et al., 2007). Only samples with [NH$_4^+$] higher than the limit of detection for the azide method of 2 µmol·L$^{-1}$ due to a significant reagent blank were analyzed for $\delta^{15}$N.   Samples were diluted to at least 10 µmol·L$^{-1}$ of NH$_4^+$ and then oxidized

to NO$_2^-$ using BrO$^-$ in an alkaline solution as previously described (Zhang et al., 2007). After a reaction time of at least 30 minutes, the reaction was stopped by 0.4 mL addition of 0.4 mol·L$^{-1}$ sodium arsenite to remove excess BrO$^-$. The concentration of the product NO$_2^-$ was then measured using a colorimetric absorption technique automated using a discrete UV-Vis spectrophotometer, to confirm the quantitative conversion of NH$_4^+$ to NO$_2^-$. The product NO$_2^-$ was reduced to nitrous oxide (N$_2$O) using sodium azide buffered in an acetic acid solution based on previously described chemical protocols

(McIlvin and Altabet, 2005).



Samples were then analyzed for their $\delta^{15}N(N_2O)$ composition using an automated $N_2O$ extraction system coupled to a

continuous flow isotope ratio mass spectrometer for *m/z* 44, 45, and 46 measurements. These measurements were conducted

at Brown University for samples collected within the US and at the Institute of Applied Ecology, Chinese Academy of

Sciences for samples collected within the Shenyang Tunnel. In each batch analysis, samples were calibrated relative to

internationally recognized N isotopic $NH_4^+$ reference materials. These reference materials underwent the same chemical

processing as the samples and were used to correct for isotopic fractionation and blank effects resulting from the chemical

conversion of $NH_4^+$ to $N_2O$. At Brown University, two international reference materials were used that included IAEA-N2

and USGS25 with $\delta^{15}N(NH_4^+)$ values of 20.3 ‰ and -30.3 ‰, respectively (Böhlke et al., 1993; Gonfiantini, 1984).

Repeated measurements of these reference materials yielded a standard deviation (±1σ) of ±0.65 ‰ (IAEA-N2; n=25) and

±0.73 ‰ (USGS25; n=25) and an overall pooled standard deviation of ±0.69 ‰ (n=50). At the Institute of Applied Ecology,

Chinese Academy of Sciences, three reference materials were used that included IAEA-N1, USGS25, and USGS26 with

$\delta^{15}N(NH_4^+)$ values of 0.4 ‰, -30.3 ‰, and 53.7 ‰ (Böhlke et al., 1993; Gonfiantini, 1984), respectively. These materials

had measured standard deviations of ±0.53 ‰ (IAEA-N1; n=8), ±0.24 ‰ (USGS25; n=8), and ±0.45 ‰ (USGS26; n=8) and

an overall pooled standard deviation of ±0.42 ‰ (n=24). All N isotopic compositions are reported relative to reference

standards using delta (δ) notation in units of per mil (‰).

$$\delta(‰) = 1000 \left( \frac{R_{sample}}{R_{ref}} - 1 \right) \tag{2}$$

where R is the ratio of the heavy to light isotope (i.e., $^{15}N/^{14}N$), for the sample and reference, respectively. Atmospheric

nitrogen ($N_2$) is the established international delta-scale reference for N isotopic composition.

**2.4 Data Analysis**

The targeted analytes were corrected for field blanks, and ambient air concentrations were then calculated based on the

volume of sampled air and reported in units of $ppb_v$ and $\mu g \cdot m^{-3}$ for $NH_3$ and $pNH_4^+$, respectively. The effective volume of

air sampled by the ALPHA passive sampler was calculated as the following:





$$V = DAt/L \tag{3}$$

where V is the volume of sampled air (m³), D is the NH₃ diffusion constant (=2.09×10⁻⁵ m² s⁻¹), A is the stationary air layer

within the sampler (=3.4636×10⁻⁴ m²), t is the time of exposure (h), and L is the cross-sectional area (= 0.006 m) (from

ALPHA Sampler User Instructions).

The method detection limit (MDL) for [NH₃] and [pNH₄⁺] determination for the active sampling technique (i.e., denuder-

filter pack) was calculated as three times the standard deviation of the field blanks. The MDL was reported in air

concentration units (e.g., ppbᵥ or μg·m⁻³) based on the typical collection times and reported separately for each sampling

environment (Table 1). The reported [NH₃] and [pNH₄⁺] precision using the denuder-filter pack sampling device was based

on five separate replicate sample collections conducted at the near-highway stationary site and expressed as the relative

standard deviation (RSD %) (Table 1). The error bars of [NH₃] and [pNH₄⁺] quantified using the denuder-filter pack in

subsequent figures represent the ±RSD % when above the MDL. Some collections had [pNH₄⁺] below the MDL, and these

samples were reported as 0.5MDL ± 0.5MDL. Multiple passive samplers (i.e., ALPHA) were always simultaneously

collected, such that RSD % was not explicitly determined, and results were reported as x̄±1σ of the multiple collections.

Significant NH₄⁺ field blanks were found on the acid-coated honeycomb denuder and the acid-coated cellulose filter. A

subset of these blanks was analyzed for δ¹⁵N(NH₄⁺) and found to have relatively consistent values of -9.6±1.3 ‰ (n=3) and -

10.9±1.4 ‰ (n=3) for the acid-coated honeycomb denuder and acid-coated cellulose filter, respectively. Corrections for

δ¹⁵N were made based on mass-balance to account for the observed blanks as previously described (Walters et al., 2019):

$$\delta^{15}N(NH_4^+)_{sample} = \frac{\delta^{15}N(NH_4^+)_{total}[NH_4^+]_{total} - \delta^{15}N(NH_4^+)_{blank}[NH_4^+]_{blank}}{[NH_4^+]_{total} - [NH_4^+]_{blank}} \tag{4}$$

Blank corrections were made for all collection media for a $f_{Blank}$ value up to 30 %, as the propagated δ¹⁵N uncertainty

generally did not exceed ±2.5 ‰ for this $f_{Blank}$ value. Samples with a $f_{Blank}$ higher than 30 % were not reported for δ¹⁵N.

This requirement as well as the azide method detection limit (i.e., 2 μmol·L⁻¹) limited our ability to quantify δ¹⁵N(pNH₄⁺) for





samples collected at the near-highway monitoring site and mobile on-road measurements, such that only $\delta^{15}N(NH_3)$ was

reported for the collections conducted in the US. The collection media blank also impacted the mobile $\delta^{15}N(NH_3)$

measurements, as 6 out of 20 samples had a $f_{Blank} > 30$ %. Error bars reported for subsequent $\delta^{15}N$ values represent the

propagated uncertainty that includes the collection uncertainty and the blank contribution. Replicate collected samples at the

near-highway site indicated that $\delta^{15}N(NH_3)$ from $NH_3$ collected using an acid-coated denuder had an average reproducibility

within 0.8 ‰ (n=5) (Table 1), consistent with previous field measurements (Walters and Hastings, 2018).

## 3. Results

**3.1 Active $NH_x$ Collection Using a Denuder-Filter Pack**

**3.1.1 Near Highway-Measurements (Providence, RI, US)**

Seasonal $NH_x$ collections at the near-highway monitoring site were performed under a variety of environmental conditions

(Table 2). Temperature and relative humidity averaged ($\bar{x}\pm1\sigma$) 23.7±3.0 ℃ and 63.3±17.4 % during summer and -0.2±4.6

℃ and 55.0±15.2 % during winter. The prevailing wind direction was from SSE to WSW during summer and from WNW

to NW during winter (Table 2). The near highway [CO], a tracer of fossil-fuel combustion, data exhibited expected diel

patterns with elevated values in early morning (6:00 to 9:00) rush hour that was observed during both summer and winter

(Figure 1a; Figure S4).

The near-highway [$NH_3$] ranged between 5.8 and 20.2 $ppb_v$ during summer and 2.4 and 20.9 $ppb_v$ during winter at the near-

highway monitoring location (Figure 1b). The greatest collection of $NH_3$ was 55.1 µg, which was well within the operative

capacity of the citric acid-coated honeycomb denuder of ~400 µg (Walters and Hastings, 2018). The average [$NH_3$] ($\bar{x}\pm1\sigma$)

was 14.0±4.0 $ppb_v$ (n=32) and 12.0±4.8 $ppb_v$ (n=22) for summer and winter, respectively (Table 2), which was not found to

be significantly different (p>0.05). Diel [$NH_3$] patterns were observed during both summer and winter, with significantly

lower values occurring during the night/early morning collection period (Table 2). The dependence of [$NH_3$] on the vector

averaged wind direction is shown in Figure 2. Overall, the near-highway monitoring site was downwind of I-95 (i.e., SSE to



N; Figure 1) for 51 out of 54 $NH_x$ collection periods (Figure 2). The [$NH_3$] was significantly lower when the wind direction indicated the monitoring site was upwind of I-95 compared to when downwind of I-95, with averages of 5.8±2.7 $ppb_v$ (n=3), and 13.6±4.2 $ppb_v$, respectively (p < 0.01). Strong positive linear correlations were found between [$NH_3$] and the mean [CO] during each collection period during summer (r = 0.736, p <0.01) and winter (r = 0.821, p<0.01), with a slope (mol:mol) of

0.025±0.005 and 0.027±0.005, respectively (Figure 4).

Roadside [$pNH_4^+$] ranged from 0.045 (i.e., 0.5MDL) to 0.938 $\mu g \cdot m^{-3}$ and from 0.117 (i.e., 0.5MDL) to 2.327 $\mu g \cdot m^{-3}$ during summer and winter, respectively (Figure 1c). The average [$pNH_4^+$] was 0.302±0.208 $\mu g \cdot m^{-3}$ (n=32) and 0.530±0.468 $\mu g \cdot m^{-3}$ (n=22) during summer and winter, respectively (Table 2), which was significantly different (p<0.05). During winter, an

[$pNH_4^+$] outlier of 2.327 $\mu g \cdot m^{-3}$ was identified based on a Grub's t-test (p<0.05). However, even with the removal of this outlier, the seasonal [$pNH_4^+$] average was found to be significantly different (p<0.05). $NH_x$ speciation was quantified as f($NH_3$) (Eq. 5).

$$f_{NH_3} = \frac{[NH_3]\ (mol)}{[NH_3 + pNH_4^+](mol)} \tag{5}$$

Overall, f($NH_3$) ranged from 0.889 to 0.996 during summer and 0.878 to 0.986 during winter (Figure 1d), indicating that

$NH_3$ was the dominant $NH_x$ species during both summer and winter. The average f($NH_3$) was 0.972±0.022 (n=32) and 0.944±0.029 (n=22) during summer and winter, respectively (Table 2). The average seasonal f($NH_3$) was found to be statistically different (p<0.05), indicating a greater extent of $NH_3$ partitioning to $pNH_4^+$ during winter. Significant correlations were observed between f($NH_3$) and relative humidity for both summer (r = -0.533, p<0.01) (Figure S5) and winter (r = -0.613, p<0.01) (Figure S5).


We note that the seasonal [$pNH_4^+$] and f($NH_3$) statistical differences at the near-highway monitoring location could have been influenced by a negative collection artifact associated with $pNH_4^+$ volatilization off the PTFE filter used during the summer near-highway sampling campaign, which would have artificially lowered our quantification of [$pNH_4^+$]. Previous




studies have reported a pNH$_4^+$ volatilization ranging from 0 to 30% in denuded-PTFE filter collections (Yu et al., 2006). It

is difficult to quantify the exact pNH$_4^+$ loss during the summertime near-highway measurements; however, assuming a

maximum of 30% pNH$_4^+$ volatilization would have reduced the average f(NH$_3$) from 0.972±0.022 to 0.964±0.027 (Table

S1).  Even when considering an upper range of pNH$_4^+$ volatilization, the summer and winter f(NH$_3$) were found to be

significantly different (p<0.05), indicating this potential sampling artifact did not impact the trends of the reported [pNH$_4^+$]

or the f(NH$_3$) results.


The measured δ$^{15}$N(NH$_3$) ranged from 2.6 to 9.3 ‰ and from 4.9 to 10.1 ‰ during the summer and winter, respectively

(Figure 1e).  The δ$^{15}$N(NH$_3$) average was 6.4±1.7 ‰ (n=32) and 8.1±1.4 ‰ (n=22) during summer and winter, respectively

(Table 2), which were significantly different (p<0.05).  The dependence of δ$^{15}$N(NH$_3$) on the vector averaged wind direction

is shown in Figure 2.  Overall, the δ$^{15}$N(NH$_3$) values were not found to be significantly different when the monitoring site

was upwind or downwind of I-95, with averages of 7.6±1.4 ‰ (n=3) and 7.1±1.8 ‰ (n=51), respectively (p>0.05).  No

statistical difference was found between the collection period and δ$^{15}$N(NH$_3$) during the winter (p>0.05), but significantly

lower δ$^{15}$N(NH$_3$) values were observed during the summer for the night/early morning sample (0:30 to 6:30) (p<0.05) (Table

2).  Significant correlations between δ$^{15}$N(NH$_3$) and f(NH$_3$) were observed for both summer (r = 0.349, p<0.05) (Figure S5)

and winter (r = 0.535, p<0.05) (Figure S6).  However, these correlations were found to be impacted by an influential f(NH$_3$)

value during the summer and winter of 0.889 and 0.878, respectively (Figure S4 and S5).  Removing these influential f(NH$_3$)

observations, resulted in an insignificant correlation between δ$^{15}$N(NH$_3$) and f(NH$_3$) for both summer (r = 0.300, p >0.05)

(Figure S4) and winter (r = 0.378, p >0.05) (Figure S5).

### 3.1.2 Tunnel Measurements in Shenyang, Liaoning, China

Tunnel temperature and relative humidity conditions remained relatively consistent throughout our sampling campaign and

averaged 19.3±1.6 ℃ and 35.4±6.7 %, respectively (Table 3).  Due to the elevated concentrations in the tunnel, the amount

of collected NH$_3$ on the acid-coated honeycomb denuder averaged 406±125 μg, indicating the laboratory determined


operative capacity of ~400 µg was often exceeded (Walters and Hastings, 2018). The citric acid-coated filter collected no more than 275 µg of NH₃, which was within the laboratory determined operative capacity of at least 350 µg (Walters et al.,

2019). Thus, our NHₓ measurements are expected to be accurate, but there could be uncertainty in the NHₓ speciation, as the acid-coated filter may represent a combination of NH₃ breakthrough and pNH₄⁺ volatilization. Therefore, our concentration results and analysis of samples collected in the Shenyang tunnel will focus on [NHₓ]. Overall, [NHₓ] was elevated ranging from 64.4 to 210.6 ppbᵥ and averaged 132.5±45.8 (n=21) (Figure 4a; Table 3). An obvious [NHₓ] diel cycle was observed in which higher concentrations occurred during periods that the tunnel was open compared to sampling periods in which the

tunnel was closed to vehicle passage, with averages of 136.8±18.8 ppbᵥ (n=7), 181.2±23.0 ppbv (n=7), and 79.4±14.4 ppbᵥ (n=7), for the 6:00 to 14:00, 14:00 to 22:00, and 22:00 to 6:00 collection periods, respectively (Table 3).

We have estimated f(NH₃), assuming that the pNH₄⁺ in PM₂.₅ was linked to the SO₄²⁻—NO₃⁻—NH₄⁺ thermodynamic system, and that the influence of other ions (e.g., Na⁺, Ca²⁺, or Cl⁻) had a negligible impact on the chemistry of this system (Shah et

al., 2018). Ion-mass balance was utilized to calculate the expected [pNH₄⁺] for each collection period based on the measured [pNO₃⁻] and [pSO₄²⁻] (Figure 4b) from the aqueous filter extracts Eq. (6):

$$[pNH_4^+](mol) = (2[pSO_4^{2-}] + [pNO_3^-])(mol) \qquad (6)$$

Utilizing the ion-mass balance approach, f(NH₃) was estimated to range between 0.856 to 0.997 and averaged 0.956±0.038 (Figure 4c; Table 3). NHₓ speciation was also estimated using ISORROPIA (Fountoukis and Nenes, 2007; Nenes et al.,

1998). Model inputs included the measured [NHₓ], [pNO₃⁻], and [pSO₄²⁻], and average relative humidity and temperature for each collection period, and the model was run in the forward direction in the meta-stable state. The f(NH₃) was then calculated based on the model output of [NH₃] and [pNH₄⁺] (Table S2). Overall, there was a near-exact agreement between the ion-mass balance and the ISORROPIA f(NH₃) estimates (though note that ISORROPIA was not used for the first five collection periods due to the absence of relative humidity and temperature data) (Figure 5c). Overall, this analysis indicated

that NHₓ in the tunnel was primarily in the form of NH₃, consistent with the near-highway stationary observations (Figure 1d), such that NHₓ dynamics can be considered as approximately representative of NH₃ dynamics.





Since NHₓ speciation was not achieved in the tunnel collections due to denuder saturation, our reported isotopic results and analysis will focus on $\delta^{15}N(NH_x)$, with the expectation that it primarily represents $NH_3$. The $\delta^{15}N(NH_x)$ was calculated for

each sampling period using mass-balance (7):

$$\delta^{15}N(NH_x) = f_{NH_4^+ - denuder}\, \delta^{15}N(NH_4^+)_{denuder} + f_{NH_4^+ - Nylon}\, \delta^{15}N(NH_4^+)_{Nylon} + f_{NH_4^+ - acid\ filter}\, \delta^{15}N(NH_4^+)_{acid\ filter}$$

(7)

where $f_{NH4+\text{-}denuder}$, $f_{NH4+\text{-}Nylon}$, and $f_{NH4+\text{-}acid\ filter}$, represents the fraction of $NH_4^+$ extracted from the denuder, Nylon filter, and acid-coated filter, respectively for each sampling event. Overall, the $\delta^{15}N(NH_x)$ ranged from -1.6 to 9.2 ‰ (Figure 4d) and

had a numerical average of 2.9±2.5 ‰ (n=21) (Table 3). There was a strong diel cycle in $\delta^{15}N(NH_x)$ in which the 22:00 to 6:00 collection period that included the period the tunnel was closed (i.e., 23:00 to 5:00) resulted in a statistically lower $\delta^{15}N(NH_x)$ of 0.1±1.3 ‰ (n=7), relative to the 6:00 to 14:00 and 14:00 to 22:00 collection periods that averaged 3.6±1.0 ‰ (n=7) and 4.8±2.0 ‰ (n=7), respectively (p<0.05) (Table 3). No significant correlation was observed between $\delta^{15}N(NH_x)$ and with the estimated $f(NH_3)$ (r = 0.383, p>0.05) (Figure S7).


### 3.1.3 Tunnel Measurements in Shenyang, Liaoning, China

We conducted a regional and spatial survey of traffic derived $NH_3$ emissions in the northeastern part of the US (Figure 5a) representing a range of environmental and driving conditions (Table 4; Figure 5b). Overall, the on-road [$NH_3$] ranged from 2.3 to 23.2 ppbᵥ and averaged 7.3±4.7 ppbᵥ (n=20) (Figure 5c). The highest [$NH_3$] were found to occur during collection

periods near urban cores that included Boston, MA, Providence, RI, New York City, NY, and Washington, D.C. (Figure 6a). The on-road [$NH_3$] was significantly correlated with [CO] (r = 0.821, p<0.01), and the linear relationship between $NH_3$ and CO had a slope ($NH_3$(mol):CO(mol)) of 0.026±0.005 (Figure 3), which was similar to the near-highway relation of 0.025±0.005 and 0.0270±0.005, observed during summer and winter, respectively. On-road [$NH_3$] was found to be significantly correlated with vehicle speed (r = -0.673, p < 0.01) (Figure S8). On-road [$pNH_4^+$] ranged from 0.047 (i.e.,

0.5MDL) to 0.710 μg·m⁻³ (Figure 5d) and averaged 0.204±0.176 μg·m⁻³ (n=20). NHₓ speciation indicated $NH_3$ was the




dominant species, consistent with our stationary observations, as f(NH$_3$) ranged from 0.800 to 0.987 (Figure 5e) and

averaged 0.934±0.050 (n=20).

On-road $\delta^{15}$N(NH$_3$) ranged from -3.0 to 10.1 ‰ (Figure 5f) and averaged 5.7±3.5 ‰ (n=14). On-road $\delta^{15}$N(NH$_3$) was not

found to be significantly correlated with f(NH$_3$) (r = 0.249, p>0.05) or average vehicle speed (r = -0.179, p>0.05) (Figure

S8). Spatial mapping of $\delta^{15}$N(NH$_3$) indicated the highest values near urban cores (Figure 6b). Each collection period was

categorized as either a trucking or highway route using the percentage of annual average daily truck traffic contributions to

annual average daily traffic (U.S. Dept of Transportation, 2013) similar to that previously described (Miller et al., 2017).

Routes on our on-road measurements with diesel trucks that comprised at least 25 % of the annual average daily traffic and

at least a yearly average of 8,500 diesel trucks per day were identified (U.S. Dept of Transportation, 2013), which were

located on rural highways typically outside of urban areas. This categorization technique was used to qualitatively identify

differences in vehicle fleet compositions during our measurements since real-time vehicle count data was not collected. Two

sampling collection periods were identified as a trucking route, including (1) from outside Harrisburg, PA to New

Smithville, PA along I-81 and I-78, and (2) from Kirkwood, PA to Colliersville, NY along I-81 and I-88. Though the

number of measurements conducted on trucking routes was limited in this case study, the average on-road $\delta^{15}$N(NH$_3$) on

highway and trucking routes were 6.9±1.9 ‰ (n=12) and -1.5±1.6 ‰ (n=2), respectively, which were found to be

significantly different (p<0.01).

### 3.2 Comparison Between Active and Passive NH$_3$ Collection

A comparison between the active and passive collection of NH$_3$ for concentration and $\delta^{15}$N(NH$_3$) characterization is

summarized in Table 5. The active collection sampling technique resulted in an [NH$_3$] of 12.0±1.2 ppb$_v$ and 127.1±12.5

ppb$_v$ over the entire winter near-highway and Shenyang tunnel sampling campaigns, respectively. These concentrations

were calculated from the total collected NH$_4^+$ over the sampling campaign divided by the total volume of collected air for

each respective campaign, and the reported uncertainty represents the RSD of the active collection technique of 9.8 %. We



note that the [NH$_3$] in the Shenyang tunnel determined using the denuder-filter pack was not measured directly but was

   calculated from the measured [NH$_x$] and estimated f(NH$_3$). The passive collection resulted in an [NH$_3$] of 11.6±1.4 ppb$_v$

   (n=4) and 124±3.6 ppb$_v$ (n=3), during winter at the near-highway monitoring location and in the Shenyang tunnel

   respectively, which was in close agreement with the active collection technique. The mass-weighted δ$^{15}$N(NH$_x$) using the

   active collection technique was 8.0±1.1 ‰ and 3.5±0.8 ‰ during winter at the near-highway monitoring location and in the

Shenyang tunnel, respectively, where the uncertainty represents the propagated error (Table 5). We note that the tunnel

   δ$^{15}$N(NH$_3$) technically represents δ$^{15}$N(NH$_x$); however, due to the elevated estimated f(NH$_3$), δ$^{15}$N(NH$_x$) ~ δ$^{15}$N(NH$_3$).

   Passive collection had an average δ$^{15}$N(NH$_3$) of -7.7±0.1 ‰ (n=4) and -11.7±0.3 ‰ (n=3) during winter at the near-highway

   monitoring location and in the Shenyang tunnel, respectively, which was found to be significantly different from the

   δ$^{15}$N(NH$_3$) measured using active collection for each sampling campaign (p<0.01). The δ$^{15}$N(NH$_3$) difference between

passive and active collection was calculated to be -15.7±1.1 ‰ and -15.2±0.9 ‰ during winter at the near-highway

   monitoring location and in the tunnel in Shenyang China, respectively (Table 5), indicating a consistent δ$^{15}$N(NH$_3$) off-set

   between the active and passive sampling collection techniques.

## 4. Discussion

### 4.1. Active NH$_x$ Collection Using a Denuder-Filter Pack

**4.1.1. NH$_x$ Concentrations**

   Elevated [NH$_3$] were observed for our stationary and on-road measurements, which have important implications for air

   quality in urban regions (e.g., Plautz, 2018; Sun et al., 2017; Wang et al., 2015). Similar [NH$_3$]:[CO] relations were

   observed at the near-highway site during summer (0.025±0.005) and winter (0.027±0.005) and from the on-road

   measurements (0.026±0.005) (Figure 3). These observed [NH$_3$]:[CO] relations were similar to previously reported values of

0.031±0.005 from on-road measurements in New Jersey and California in the United States using high-resolution open-

   pathway sensors (Sun et al., 2014, 2017), and 0.031 to 0.038 based off fitted NH$_x$ and CO slopes from aircraft measurements

   in the California South Coast Air Basin (Nowak et al., 2012). The similarity of these measurements indicated that the traffic

   plumes measured in this study were representative of previous literature reports in the US, and the active collection of NH$_3$



using a denuder-filter pack sampling technique was suitable for reproducing accurate [NH$_3$] under traffic plume

environmental conditions.

Speciation measurements indicated that NH$_3$ was the dominant NH$_x$ component at all sampling locations as the average

f(NH$_3$) was observed to be higher than 0.934 at the varying sampling locations. The observed elevated f(NH$_3$) values were

likely driven by the limited availability of acidic species, as observed for the [pSO$_4^{2-}$] and [pNO$_3^-$] measurements that were

low relative to [NH$_x$] in the Shenyang tunnel (Figure 4c). A significant difference in seasonal [pNH$_4^+$] and f(NH$_3$) were

observed at the near-highway monitoring site, which likely reflects influences from environmental conditions during winter

that favored NH$_3$ partitioning to pNH$_4^+$. NH$_3$ and HNO$_3$ exist in thermodynamic equilibrium with lower temperatures and

higher relative humidity conditions favoring the condensed components (Behera et al., 2013):

$$NH_3 + HNO_3 \rightleftharpoons NH_4NO_3(s) \ or \ (NH_4^+(aq) + NO_3^-(aq)) \tag{8}$$

Wintertime conditions at the near-highway monitoring location had much lower temperatures relative to summertime with

an average of -0.2±4.6 °C and 23.7±3.0 °C, respectively (Table 2), which would have facilitated the formation of NH$_4$NO$_3$,

driven by the availability of the limiting reagent (i.e., HNO$_3$). However, it remains unclear whether the pNH$_4^+$ collected at

the near-highway derived from NH$_3$ vehicle emissions or long-range transport due to the extended atmospheric lifetime of

pNH$_4^+$ on the order of a few days to a week (Paulot et al., 2016). Potentially δ$^{15}$N(pNH$_4^+$) measurements could be useful in

tracking the source of NH$_3$ that contributed to the collected pNH$_4^+$, but were not available in this study due to pNH$_4^+$ mass

limitations.

**4.1.2 Traffic-Plume δ$^{15}$N(NH$_3$) Variability**

Here we assess the drivers behind the δ$^{15}$N(NH$_3$) variabilities measured within each sampling campaign including the seasonal

difference measured at the near-highway monitoring site, the temporal variation observed during summer at the near-highway

site and the Shenyang tunnel, and the spatial patterns observed from the on-road measurements. We hypothesize that the




observed $\delta^{15}N(NH_3)$ variabilities could be related to (1) $f(NH_3)$ partitioning, (2) $NH_3$ dry deposition, (3) background $NH_3$ contributions and/or (4) vehicle fleet composition differences.

Previously, it has been theoretically estimated and shown from field observations and laboratory studies that isotopic N equilibrium and reactions between $NH_3$ and $NH_4^+$ can scramble the $^{14}N$ and $^{15}N$ distributions between these molecules, leading to the preferential partitioning of $^{15}N$ into $NH_4^+$ (Kawashima and Ono, 2019; Savard et al., 2017; Urey, 1947; Walters et al., 2018). A significant positive correlation between $f(NH_3)$ and $\delta^{15}N(NH_3)$ during both summer and winter were observed at the near-highway monitoring location, which is consistent with influences from N isotopic equilibrium reactions. However, the $\delta^{15}N(NH_3)$ and $f(NH_3)$ relations were affected by a single influential $f(NH_3)$ value during both summer and winter, and removal

of these points resulted in an insignificant relation between $\delta^{15}N(NH_3)$ and $f(NH_3)$ (Figures S5 and S6). The temporal tunnel variability is not likely to be driven by $f(NH_3)$ partitioning influences as $f(NH_3)$ was not found to be significantly different between periods the tunnel was open or closed ($p>0.05$), indicating a significant change in $NH_3/pNH_4^+$ partitioning did not occur during these periods. Additionally, a significant relation between $\delta^{15}N(NH_3)$ and $f(NH_3)$ was not observed for the on-road measurements (Fig. S8). Thus, we do not expect $f(NH_3)$ partitioning and $NH_3$ reactive sink to have played a significant

role in the $\delta^{15}N(NH_3)$ variability observed at the various sampling sites. We note that the influence of N isotopic exchange reactions on $\delta^{15}N(NH_3)$ depends on the degree of $NH_3/pNH_4^+$ partitioning. Typically, $f(NH_3)$ was observed to be $>0.934$, which would limit the influence of equilibrium exchange reactions to alter the measured $\delta^{15}N(NH_3)$ values. We also note that there is an equilibration time needed before N isotopic equilibrium between $NH_3$ and $pNH_4^+$ is achieved, but this rate is currently unknown. Thermodynamic gas-fine aerosol equilibrium has been calculated to have an equilibration time on the

order of 10s of minutes to several hours, dependent upon ambient conditions and particle characteristics (Meng and Seinfeld, 1996). Assuming a similar equilibration rate for N isotopic exchange between $NH_3$ and $pNH_4^+$ would indicate that complete N isotopic equilibrium would likely not be achieved in close proximity to $NH_3$ emission sources, which is consistent with our observations.





NH$_3$ dry deposition was not expected to contribute to the observed variability in the well-ventilated sampling conditions at the near-highway monitoring location and the on-road measurements.    These measurements were conducted in close proximity to the emitted NH$_3$ (e.g., typically within 5 m at the near-highway monitoring site), which should have minimized NH$_3$ loss via dry deposition (Asman et al., 1998). However, NH$_3$ dry deposition may have played an important role under the closed sampling environment of the tunnel and may explain the observed $\delta^{15}$N(NH$_x$) temporal variability with higher values observed

when the tunnel was open (4.2±1.7 ‰, n=14) compared to samples collected during periods that the tunnel was closed to traffic (0.1±1.3 ‰, n=7) (Table 3).   Previously, lower NH$_3$ emission ratios were reported from traffic plumes in tunnels relative to on-road highway measurements, which was concluded to result from contributions of NH$_3$ dry deposition on the tunnel surfaces (Sun et al., 2017). If NH$_3$ dry deposition is influenced by N isotopic equilibrium reactions between NH$_3$ and the surface deposited NH$_4^+$, this would have resulted in NH$_3$ depleted in $^{15}$N as it is removed from the atmosphere resulting in lower

$\delta^{15}$N(NH$_3$) values (Walters et al., 2018).   Indeed, a previous NH$_3$ absorption-desorption study on minerals has shown the preferential removal of $^{15}$NH$_3$ from the gaseous phase, with the degree of $^{15}$N depletion of the gaseous NH$_3$ dependent upon the adsorbed NH$_3$ amount (Sugahara et al., 2017).   Thus, as the traffic plume ages in the absence of fresh emissions, we would expect NH$_3$ dry deposition influences and the potential for N isotopic exchange reactions between the air and tunnel surface to be most significant, which might explain the lower $\delta^{15}$N(NH$_x$) values observed during periods the tunnel was closed.   Dry

deposition of NH$_3$ during the day in the tunnel could have also impacted the measured $\delta^{15}$N(NH$_3$) values, but the constant emission of NH$_3$ likely resulted in non-equilibrium conditions, such that N isotopic equilibrium between the ambient NH$_3$ and surface deposited NH$_4^+$ would not have been fully achieved.

Background NH$_3$ contributions are important to identify as a possible driver of $\delta^{15}$N(NH$_3$) variability.   At the near-highway

monitoring site, wind sector analysis found no statistical difference in $\delta^{15}$N(NH$_3$) when sorted by wind direction for either summer or winter (Figure 2).   This indicates that transport from local NH$_3$ point sources other than vehicle emissions played a minor role in the seasonal $\delta^{15}$N(NH$_3$) difference.   Additionally, the similar [NH$_3$]:[CO] seasonal relations at the near-highway monitoring site (Figure 3) indicates that seasonal variations in background NH$_3$ influences at the near-highway monitoring site were minor.   While dilution by background air into the Shenyang tunnel during the periods that the tunnel



was closed to traffic should be considered as a driver of the temporal $\delta^{15}N(NH_x)$ variability, both the average [$NH_x$] and

f($NH_3$) were not consistent with significant mixing in of background air.  When the tunnel was closed [$NH_x$] averaged

79.4±14.4 ppb$_v$ (Table 3), which is elevated compared to urban background [$NH_3$] measurements previously reported from a

megacity in China (Beijing) during winter of 5.22±3.75 µg·m$^{-3}$ (or 6.9±4.9 ppb$_v$) (Ianniello et al., 2010).  Additionally,

f($NH_3$) was elevated during the collection period that the tunnel was closed, averaging 0.937±0.045 (Table 3), consistent

with local emissions rather than contributions from background air that tends to have a lower f($NH_3$) value such as reported

to be typically below 0.6 during November based on data collected from Beijing, China (Zhang et al., 2018).  Thus, we do

not expect background $NH_3$ contribution to have played a significant role in the tunnel temporal $\delta^{15}N(NH_x)$ variability.

Background $NH_3$ contributions could have played a role on the on-road spatial $\delta^{15}N(NH_3)$ variability as high-resolution

satellite measurements have indicated an elevated $NH_3$ column enhancement centered around Lancaster, PA, US (Van

Damme et al., 2018), which is near the trucking route that resulted in the lowest $\delta^{15}N(NH_3)$ of -3.0±1.8 ‰.  The measured

[$NH_3$] along this route was not exceptionally elevated (6.2 ppb$_v$), but we could not compare its [$NH_3$]:[CO] relation as a

tracer of potential mobile emissions because the CO analyzer was off during this collection period.  The $NH_3$ source of this

column enhancement is suggested to be related to agricultural activities in southeast PA, US, which tends to have a low

$\delta^{15}N(NH_3)$ signature (e.g., -31 to -14 ‰; Hristov et al., 2011). This $\delta^{15}N(NH_3)$ signature is much lower than the vehicle

emitted value found near urban regions and could have influenced the lower $\delta^{15}N(NH_3)$ value for the on-road measurement

found near Lancaster, PA, US.  However, $NH_3$ volatilization is highly temperature-dependent (He et al., 1999), and we

expect these emissions to be minimal during the winter when the on-road measurements were conducted.  We note that in

addition to the elevated $NH_3$ column enhancement near Lancaster, PA, US there are pockets of elevated $NH_3$ concentrations

detected via high-resolution satellite observations near Hagerstown, MD, US (Van Damme et al., 2018); however, we did not

observe a characteristically low $\delta^{15}N(NH_3)$ value along this route.  Additionally, there does not appear to be a pocket of

elevated $NH_3$ concentrations along the route that resulted in the second-lowest $\delta^{15}N(NH_3)$ of 0.1±2.6 ‰, which was observed

on the trucking route from Kirkwood, PA, US to Colliersville, NY, US.  Therefore, we do not expect contributions from





background NH$_3$ to have played an important role in explaining the observed spatial δ$^{15}$N(NH$_3$) variabilities in the on-road

measurements.

Vehicle fleet compositions could have a strong influence on the measured δ$^{15}$N(NH$_3$) variabilities if gasoline and diesel-powered engines, which utilize different types of NO$_x$ reduction technologies that lead to NH$_3$ emission (Suarez-Bertoa and Astorga, 2018), have different δ$^{15}$N(NH$_3$) emission signatures.  Categorization of our on-road collection routes as either

highway routes or trucking routes resulted in statistically significantly different δ$^{15}$N(NH$_3$) values of 6.9±1.9 ‰ (n=12) and -1.5±1.6 ‰ (n=2), respectively, supporting the idea that the δ$^{15}$N(NH$_3$) spatial variation was influenced by fleet composition. This would also be consistent with previous findings that vehicle fleet composition was the main driver of spatial on-road variability observed for δ$^{15}$N(NO$_x$) (Miller et al., 2017).  Vehicle fleet NH$_3$ emissions driven by reduction technologies may have also influenced the seasonal δ$^{15}$N(NH$_3$) difference observed at the near-highway monitoring location.  Under cold ambient

conditions of -7 ℃, diesel-powered vehicles equipped with SCR technology were reported to have minimal emission of NH$_3$ (below the measurement limit of detection), while gasoline-powered vehicles equipped with TWCC were reported to have increased NH$_3$ emission relative to warmer conditions at 23 ℃ (Suarez-Bertoa and Astorga, 2018).  Vehicle fleet composition may also explain the statistically significantly lower δ$^{15}$N(NH$_3$) values during the summer night/early morning collection period at the near-highway monitoring site (Table 2).  Vehicle fleet composition was not monitored in this study, but a previous

study has reported relatively higher truck traffic compared to gasoline vehicles from near-highway measurements during the night/early morning before morning rush hour (Wang et al., 2018).  A lower δ$^{15}$N(NH$_3$) signature from diesel emissions compared to gasoline, as supported by our on-road measurements, would explain both the seasonal differences in δ$^{15}$N(NH$_3$) and the temporal δ$^{15}$N(NH$_3$) variability observed primarily during summer.  To date, there are neither direct tailpipe measurements of δ$^{15}$N(NH$_3$) from gasoline and diesel-powered vehicle nor an explanation for the observed or expected

δ$^{15}$N(NH$_3$) signatures of vehicle derived emissions. Future work is needed to evaluate direct tailpipe δ$^{15}$N(NH$_3$) signatures from gasoline and diesel-powered vehicles to test our hypothesis.  We note that while there was a statistically significant seasonal difference in the measured δ$^{15}$N(NH$_3$) at the near-highway monitoring site, the absolute difference of ~1.7‰ was quite small.





## 4.2. Comparison Between Active and Passive NH$_3$ Collection

A comparison between active and passive sampling was conducted to evaluate the performance of the varying NH$_3$
collection techniques. Overall, remarkably similar [NH$_3$] were determined using the active (i.e., denuder-filter pack) and
passive (i.e., ALPHA) sampling techniques (Table 5). The [NH$_3$] determined from the active sampling technique over the
entire campaign was 12.0±1.2 ppb$_v$ and 127.1±12.5 ppb$_v$, which compared with the passive sampling technique with values
of 11.6±1.2 ppb$_v$ and 124±3.6 ppb$_v$ at the near-highway and in the Shenyang tunnel, respectively (Table 5). The finding of
similar [NH$_3$] between passive and active sampling of NH$_3$ is consistent with previous comparisons (Day et al., 2012), and
provides support that passive collection of NH$_3$ may be a useful approach for spatial documentation of near-surface [NH$_3$].

While the two sampling techniques produced consistent [NH$_3$], significant differences in $\delta^{15}$N(NH$_3$) were observed. The
mass-weighted $\delta^{15}$N(NH$_3$) using the active sampling technique was 8.0±1.1 ‰ and 3.5±0.8 ‰, while the values using the
passive sampling technique was -7.7±0.1 ‰ (n=4) and -11.7±0.3 ‰ (n=3) at the near-highway site (winter) and in the
Shenyang tunnel, respectively (Table 5). The measured traffic derived $\delta^{15}$N(NH$_3$) values via passive sampler were similar to
previous measurements utilizing a similar sampling approach that included measurements in a tunnel in the US and a tunnel
in China with reported values of -3.4±1.2 ‰ (n=2) (Felix et al., 2013) and -14.2±2.6 ‰ (n=8) (Chang et al., 2016),
respectively. While our passive $\delta^{15}$N(NH$_3$) values were generally consistent with previous reports, there is a large off-set
between the passive and active sampling techniques that were calculated to be -15.7±1.1 ‰ and -15.2±0.9 ‰ at the near-
highway site and in the Shenyang tunnel, respectively (Table 5). This consistent $\delta^{15}$N(NH$_3$) bias indicates that the passive
samplers were precise but not accurate in their $\delta^{15}$N(NH$_3$) characterization.

The passive samplers are limited by diffusion, as demonstrated in Eq. (3), which was used to calculate [NH$_3$] concentrations
using this sampling technique. Thus, we hypothesize that a diffusion isotope effect may impact $\delta^{15}$N(NH3) for passive
sampling in ambient environments. This effect is driven by relative mass differences, which has been previously estimated





to be -28 ‰ (Pan et al., 2016), suggesting that the lighter $^{14}NH_3$ would preferentially diffuse to the passive sampler pad

leading to a potential offset of -28 ‰ between the collected $\delta^{15}N(NH_3)$ relative to the ambient $\delta^{15}N(NH_3)$ in open sampling

conditions.  We note that the previously estimated diffusion isotope effect (e.g., Pan et al., 2016) does not consider the

diffusion of the isotopic species in gases (i.e., air), but rather the calculation was based off an effusion process, which is

more typical of low-pressure behavior. We estimate the $NH_3$ diffusion fractionation factor ($\alpha_{diff}$) as the relative diffusion

rates (D) of $^{14}NH_3$ and $^{15}NH_3$ through air (molar mass ~28.92 g) as the square root of the inverse of the reduced masses ($\mu$) of

the $^{14}NH_3$ and $^{15}NH_3$ isotopologues with air (mass~28.9 g mol$^{-1}$):

$$\alpha_{diff} = \frac{D_{^{15}NH_3}}{D_{^{14}NH_3}} = \sqrt{\frac{\mu_{^{14}NH_3-air}}{\mu_{^{15}NH_3-air}}} = 0.982 \tag{9}$$

Eq. (9) indicates $\alpha_{diff}$ for the N isotopologues of $NH_3$ to be ~0.982, resulting in an enrichment factor ($\varepsilon$(‰)=1000($\alpha$-1)) of -

17.7‰.  Our estimated diffusion fractionation effect is quite near the observed $\delta^{15}N(NH_3)$ off-set between passive and active

sampling of approximately -15.5±1.0 ‰.

Overall, the large $\delta^{15}N(NH_3)$ off-set observed between passive and active $NH_3$ collection and potential $\delta^{15}N(NH_3)$ bias in the

passive collection of $NH_3$ has several important implications.  The majority of reported $\delta^{15}N(NH_3)$ source signatures have

been characterized using passive sampling techniques and might be biased by approximately -15.5 ‰ under the

environmental conditions during our sampling periods.  These previous measurements could potentially be corrected, but the

further characterization of the passive sampler $\delta^{15}N(NH_3)$ off-set is needed.


### 4.3.  Urban Traffic Plume $\delta^{15}N(NH_3)$ Signature and Implications

The measured $\delta^{15}N(NH_3)$ traffic plume signatures utilizing active sampling technique demonstrates an overall range from -3.0

to 10.1 ‰ (Figure 7).   Our analysis indicated that $\delta^{15}N(NH_3)$ variability was influenced by fleet composition and $NH_3$ dry

deposition in aged vehicle plumes measured in a tunnel.  Thus, for deriving an urban traffic plume $\delta^{15}N(NH_3)$ signature, we





have considered measurements conducted under fresh plume conditions and on/near highway measurements, as representative

of urban vehicle $NH_3$ emissions. These observations included the near-highway measurements conducted during both summer

and winter, the mobile on-road measurements conducted on highways, and the Shenyang tunnel during operation. While there

are $\delta^{15}N(NH_3)$ differences between sampling environments for this subset of observations (Figure 7), the absolute difference

in the mean $\delta^{15}N(NH_3)$ was quite small (generally within ~3 ‰) and may reflect actual differences in urban vehicle fleet

compositions. Overall, the constrained observations assumed to be representative of urban vehicle emissions reduces the

$\delta^{15}N(NH_3)$ variability with a range of 2.1 to 10.1 ‰ (Figure 7). The constrained $\delta^{15}N(NH_3)$ has a combined numerical average

of 6.6±2.1 ‰ (n=80) (Figure 7), which was found to not significantly differ from a normal distribution (Kolmogorov-Smirnov

test of normality, p = 0.528), and is suggested to be the urban vehicle-derived traffic plume $\delta^{15}N(NH_3)$ source signature.

The recommended $\delta^{15}N(NH_3)$ vehicle-derived traffic signature of 6.6±2.1 ‰ (n=80) has a narrower range and higher value

than previously reported vehicle signatures of -17.8 to 0.4 ‰ (Chang et al., 2016; Felix et al., 2013; Smirnoff et al., 2012).

The difference between the recommended $\delta^{15}N(NH_3)$ vehicle-derived source signature and previous reports by Chang et al.,

2016 and Felix et al., 2013, was found to be caused by a $\delta^{15}N(NH_3)$ bias from passive $NH_3$ collection that was suggested to be

driven by a diffusion isotope effect. The recommended $\delta^{15}N(NH_3)$ vehicle-derived source signature was also found to be

statistically different from a previous report that actively sampled $NH_3$ using a filter pack collection system, which reported

an average $\delta^{15}N(NH_3)$ of -2.1±1.9 ‰ (Smironff et al., 2012). Differences between our recommended $\delta^{15}N(NH_3)$ value and

previous reports by Smirnoff et al., 2012 are difficult to identify and may be related to differences in vehicle fleet compositions.

Additionally, we note that this difference may be related to the potential for a positive sampling artifact associated with filter

pack collection using a particulate filter and subsequent acid-coated filter for separate $pNH_4^+$ and $NH_3$ collection, respectively,

as volatilization of the collected $pNH_4^+$ could have resulted in a $NH_3$ collection bias (Yu et al., 2006). Indeed, previous

laboratory experiments have shown that $NH_3$ volatilized from $NH_4NO_3$ particles collected from filters have a $\delta^{15}N(NH_3)$ value

lower than the $\delta^{15}N(pNH_4^+)$ by 28.6±2.7 ‰ (Walters et al., 2019). Thus, $pNH_4^+$ volatilization could have artificially lowered

the reported $\delta^{15}N(NH_3)$ value and may explain the lower $\delta^{15}N(NH_3)$ values reported in Smirnoff et al. 2014 compared to our

results.


The recommended $\delta^{15}N(NH_3)$ vehicle-derived signature of 6.6±2.1 ‰ has important implications for urban air quality, as it is the only source signature that we are aware of that has a positive value (Chang et al., 2016; Felix et al., 2013; Freyer, 1978; Heaton, 1987; Hristov et al., 2011). For example, previous works have reported $\delta^{15}N(NH_3)$ values of -31 to -4.4 ‰ for animal waste $NH_3$ volatilization (Freyer, 1978; Heaton, 1987 Hristov et al., 2011), -14.6 and -11.3 ‰ for fuel combustion (coal power

plant with SCR technology) (Felix et al., 2013), and -20.1 ‰ for industrial processes (steel coke plant) (Heaton, 1987), utilizing similar type of active sampling or complete $NH_3$ capture techniques. Therefore, $\delta^{15}N(NH_3)$ is a promising technique to track vehicle derived emissions and evaluate its role as on urban budgets and N deposition patterns.

## 5. Conclusions

We characterized the $\delta^{15}N(NH_3)$ signatures from a variety of temporal and spatial traffic derived plumes utilizing a laboratory-verified active collection technique demonstrated to reflect accurate $\delta^{15}N(NH_3)$ values. Overall, our measurements indicate a $\delta^{15}N(NH_3)$ range of -3.0 to 10.1 ‰ from vehicle-derived plumes representing a variety of driving conditions and fleet compositions that included stationary measurements conducted in Providence, RI, US, and Shenyang, Liaoning, China, and mobile on-road measurements performed in the northeastern US. These $\delta^{15}N(NH_3)$ values were found to be higher than

previous reports of traffic derived measurements that ranged between -17.8 to 0.4 ‰. Our results indicate that the majority of these previously reported lower values were due to a $\delta^{15}N(NH_3)$ collection bias of approximately -15.5 ‰ associated with passive $NH_3$ collection, highlighting the critical need to utilize accurate $\delta^{15}N(NH_3)$ collection techniques.

Significant spatial and temporal $\delta^{15}N(NH_3)$ variabilities were observed in the seasonal and summer diel measurements

conducted at the near-highway monitoring site, in aged traffic plumes in the Shenyang tunnel, and along rural trucking routes in the northeastern US. Vehicle fleet composition was suggested to drive significant $\delta^{15}N(NH_3)$ variability, as suspected higher diesel $NH_3$ emissions during summer relative to winter and mobile measurements conducted on trucking routes were found to result lower $\delta^{15}(NH_3)$ values, which likely reflects differences in $NH_3$ production via three-way catalytic converter and selective catalytic reduction technologies. Additionally, physical processing associated with $NH_3$ dry deposition was suspected to have

lowered the observed $\delta^{15}N(NH_3)$ values in the tunnel when vehicle passage was ceased. The reactive $NH_3$ sink associated with

pNH$_4^+$ formation was found to play a minor role in the $\delta^{15}N(NH_3)$ variability due to elevated f($NH_3$). Accounting for these

influences, our results constrain the $\delta^{15}N(NH_3)$ signature from urban traffic derived fresh plume emissions to 6.6±2.1 ‰ ($\bar{x}$±1σ;

n = 80). In addition to $\delta^{15}N(NH_3)$ characterization, our measurements demonstrate elevated $NH_3$ emissions from vehicle

plumes and a strong relationship between [NH$_3$]:[CO] (mol:mol) with fitted slopes of 0.025±0.005, 0.027±0.005, and

0.026±0.005 for summer near-highway, winter near-highway, and on-road measurements, respectively, which are in agreement

with recent measurements in other regions. Overall, our results highlight the significance of traffic derived $NH_3$ emissions and

demonstrates the potential to use $\delta^{15}N(NH_3)$ to track its contributions to chemistry and N deposition budgets.

The results of this study have important implications for evaluating $NH_3$ budgets, particularly in urban regions. The measured

$\delta^{15}N(NH_3)$ traffic signature (6.6±2.1 ‰, n=80) is unique as it is the only source that has a reported positive $\delta^{15}N(NH_3)$ value.

Thus, $\delta^{15}N(NH_3)$ may be a useful tracer to evaluate the contribution of traffic derived emissions in urban regions and to evaluate

the connection between urban $NH_3$ emissions and its role in PM$_{2.5}$ formation. Our demonstrated approach for utilizing a

laboratory-verified technique with potential for hourly time resolution is applicable for constraining other important $NH_3$

emissions sources to produce a consistent database of $\delta^{15}N(NH_3)$ source signature values. Future work is needed to accurately

characterize and improve upon the $\delta^{15}N(NH_3)$ source inventory and evaluate potential fractionation influences associated with

NH$_x$ plume aging and deposition.

**Data availability.** Data presented in this manuscript are available on the Brown University Digital Repository at

https://doi.org/10.26300/q3h4-7s93, and the RI-DEM monitoring data are publicly available via the U.S. EPA Air Quality

System Data Mart at https://aqs.epa.gov/api.

**Author Contribution.** WWW, LS, JC, YF, MGH designed varying aspects of the field sampling plan. WWW, LS, JC, NC

were involved in carrying out the field measurements. WWW and LS conducted all laboratory analyses of data. WWW

prepared the manuscript with contributions from all co-authors.




**Competing interests.** The authors declare that they have no conflict of interest.

**Acknowledgements.** WWW acknowledges support from an Atmospheric and Geospace Sciences National Science
Foundation Postdoctoral Fellow (Grant # 1624618) during this study. YF was supported by the grants from National Key
R&D Program of China (Grant No. 2017YFC0212704) and National Research Program for Key Issues in Air Pollution Control
(grant number DQGG0105-02). This research was also supported by funding to MGH from the National Science Foundation
(AGS 1351932). The authors also acknowledge support from an Institute at Brown for Environment and Society Internal
Seed Grant (GR300123). We thank Ruby Ho, Joseph Orchardo, Yihang Duan and many others for sampling and laboratory
assistance. We are grateful to Paul Theroux of RI-DEM/RI-DOH for access and support at the RI-DEM air monitoring site
and for providing data from these sites for our analyses.

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



**Tables**

**Table 1.** Summary of method detection limit (MDL), pooled relative standard deviations (RSD), and $\delta^{15}N$ reproducibility of $NH_3$ determined from active sampling using a denuder-filter pack (ChemComb Speciation Cartridge). The MDL is reported in units of $ppb_v$ for $NH_3$ and $\mu g \cdot m^{-3}$ for $pNH_4^+$. The MDL is reported for each sampling environment including the near-highway monitoring location in Providence, RI, US during the summer (Summer-NH) and winter (Winter-NH), on-road mobile measurements in the northeastern US (Mobile), and the Tunnel in Shenyang, Liaoning, China (Tunnel).

| Species | MDL ($ppb_v$ or $\mu g \cdot m^{-3}$) | | | | RSD(%) | $\delta^{15}N$- Reproducibility |
| | Summer- NH | Winter-NH | Mobile | Tunnel | | |
|---|---|---|---|---|---|---|
| **Active Sampling (denuder-filter pack)** | | | | | | |
| $NH_3$ | 0.088 | 0.147 | 0.415 | 0.170 | 9.8 | 0.8 ‰ |
| $pNH_4^+$ | 0.090 | 0.234 | 0.093 | 0.118 | 8.5 | N/A[a] |

[a]Separate measurement of $\delta^{15}N(pNH_4^+)$ was not conducted due to sample mass limitations.





**Table 2.** Summary of the near-highway (Providence, RI, US) environmental conditions including temperature (Temp), relative humidity (RH), wind direction and $NH_x$ data including $[NH_3]$, $[pNH_4^+]$, $f(NH_3)$, and $\delta^{15}N(NH_3)$ sorted by $NH_x$ collection period for both summer and winter. Data are reported as $\bar{x}(\pm 1\sigma)$ for each collection period during summer and winter, respectively. The number of collections made during each collection period (n) is indicated.

| Collection Period (n) | Temp (°C) | RH (%) | Prevailing Wind Direction | $[NH_3]$ (ppb$_v$) | $[pNH_4^+]$ ($\mu g \cdot m^{-3}$) | $f(NH_3)$ | $\delta^{15}N(NH_3)$ (‰) |
|---|---|---|---|---|---|---|---|
| **Summer (Aug 9 to Aug 18)** | | | | | | | |
| 0:30-6:30 (7) | 20.1(1.0) | 80.5(11.1) | WSW | 9.8(3.7) | 0.350(0.269) | 0.956(0.032) | 4.2(1.0) |
| 6:30-12:30 (8) | 24.0(1.9) | 63.6(10.2) | S | 13.4(3.7) | 0.301(0.221) | 0.973(0.016) | 7.3(1.5) |
| 12:30-18:30 (8) | 27.4(2.0) | 45.1(14.1) | SSE | 16.0(3.3) | 0.252(0.135) | 0.980(0.012) | 7.1(1.5) |
| 18:30-0:30 (9) | 23.1(1.3) | 65.8(13.5) | SSW | 15.9(1.8) | 0.310(0.183) | 0.976(0.015) | 6.9(0.7) |
| *Overall (32)* | *23.7(3.0)* | *63.3(17.4)* | *SSW* | *14.0(4.0)* | *0.302(0.208)* | *0.972(0.022)* | *6.4(1.7)* |
| **Winter (Jan 21 to Feb 1)** | | | | | | | |
| 0:00-6:00 (5) | -3.7(2.8) | 59.1(10.8) | WNW | 6.3(1.7) | 0.388(0.173) | 0.925(0.017) | 8.5(0.3) |
| 6:00-12:00 (5) | -0.8(4.3) | 53.6(13.8) | WNW | 13.4(1.5) | 0.601(0.289) | 0.947(0.024) | 8.8(1.0) |
| 12:00-18:00 (5) | 2.7(4.3) | 43.7(12.5) | WNW | 16.0(2.7) | 0.447(0.191) | 0.963(0.015) | 7.8(1.5) |
| 18:00-0:00 (7) | 0.8(4.3) | 61.2(16.0) | NW | 12.3(5.2) | 0.640(0.739) | 0.942(0.036) | 7.7(1.7) |
| *Overall (22)* | *-0.2(4.6)* | *55.0(15.2)* | *WNW* | *12.0(4.8)* | *0.530(0.468)* | *0.944(0.029)* | *8.1(1.4)* |

960

965

970





**Table 3.** Summary of the Shenyang, Liaoning, China tunnel data including temperature (Temp), relative humidity (RH), $[NH_x]$, $f(NH_3)$, $[NH_3]$, and $\delta^{15}N(NH_x)$. Data are reported as $\bar{x}(\pm 1\sigma)$ for each collection period and the overall monitoring period during summer and winter, respectively. The number of collections made during each collection period (n) is indicated.

| Collection Period | Temp | | $[NH_x]$ | | $\delta^{15}N(NH_x)^b$ |
|---|---|---|---|---|---|
| (n) | (°C) | RH (%) | (ppb$_v$) | $f(NH_3)^a$ | (‰) |
| 6:00 – 14:00 (7) | 19.2(1.1) | 35.2(4.8) | 136.8(18.8) | 0.959(0.027) | 3.6(1.0) |
| 14:00 – 22:00 (7) | 20.5(1.9) | 36.2(6.4) | 181.2(23.0) | 0.973(0.028) | 4.8(2.0) |
| 22:00 – 6:00 (7) | 18.3(0.9) | 34.7(8.1) | 79.4(14.4) | 0.937(0.045) | 0.1(1.3) |
| *Overall (21)* | *19.3(1.6)* | *35.4(6.7)* | *132.5(45.8)* | *0.956(0.038)* | *2.9(2.5)* |

$^a$f($NH_3$) was calculated from $[pNH_4^+]$ estimated using ion-mass balance based on the $[NH_x]$, $[pNO_3^-]$, and $[pSO_4^{2-}]$ measurements (see Eq. 6)

$^b$Due to the elevated f($NH_3$), the measured $\delta^{15}N(NH_x) \sim \delta^{15}N(NH_3)$.



**Table 4.** Summary of the mobile on-road measurements, including average temperature (Temp), relative humidity (RH), vehicle speed, and elevation sorted by the five measurement periods. Data are reported as x̄(±1σ) for each collection period.


| Mobile Measurement Period | Temp (ºC) | RH (%) | Vehicle Speed (km/h) | Elevation (m a.s.l.) |
|---|---|---|---|---|
| Feb 20 10:40 – Feb 20 13:40 | 16.5 (1.2) | 76.7(4.6) | 97.0(21.3) | 59.5(34.3) |
| Feb 21 10:10 – Feb 21 16:53 | 13.6 (4.4) | 77.3(9.0) | 84.3(28.2) | 40.3(30.3) |
| Feb 22 19:50 – Feb 23 00:10 | 7.8 (0.7) | 84.1(6.9) | 89.7(21.8) | 35.7(24.0) |
| Feb 23 14:20 – Feb 23 20:22 | 9.0 (2.0) | 84.6(4.0) | 87.6(26.6) | 147.8(61.9) |
| Feb 24 10:30 – Feb 24 15:55 | 10.1 (2.1) | 62.0(7.6) | 99.3(23.0) | 235.0(137.2) |









**Table 5.** Summary of [NH$_3$] and $\delta^{15}$N(NH$_3$) from the passive and active collection of NH$_3$ at the winter near-highway and Shenyang stationary monitoring locations.


| Location | [NH$_3$] (ppb$_v$) | | $\delta^{15}$N(NH$_{3/x}$) (‰) | | |
|---|---|---|---|---|---|
| | Passive[a] | Active[b] | Passive[a] | Active[c] | Shift[d] 1035 |
| Winter Near-Highway | 11.6±1.4 | 12.0±1.2 | -7.7±0.1 | 8.0±1.1 | -15.7±1.1 |
| Shenyang Tunnel | 124±3.6 | 127.1±12.5 | -11.7±0.3 | 3.5±0.8[e] | -15.2±0.9 |

[a] Values reported as x̄±1σ for 4 replicates at the near highway and 3 replicates at the Shenyang Tunnel.

[b] Values are reported as the [NH$_3$] over the entire sampling campaign and the uncertainty represents the RSD(%) of the active collection system (±9.8%).

[c] Values are reported as the mass-weighted $\delta^{15}$N(NH$_3$) value observed at the two locations and the uncertainty represents the propagated error.

[d] Calculated as the $\delta^{15}$N difference between passive and active NH$_3$ collection. The uncertainty represents the propagated error

between these two measurements.

[e] The Shenyang Tunnel active measurements represent $\delta^{15}$N(NH$_x$); however, due to elevated f(NH$_3$) that averaged, $\delta^{15}$N(NH$_x$) $\sim \delta^{15}$N(NH$_3$).







# Figures

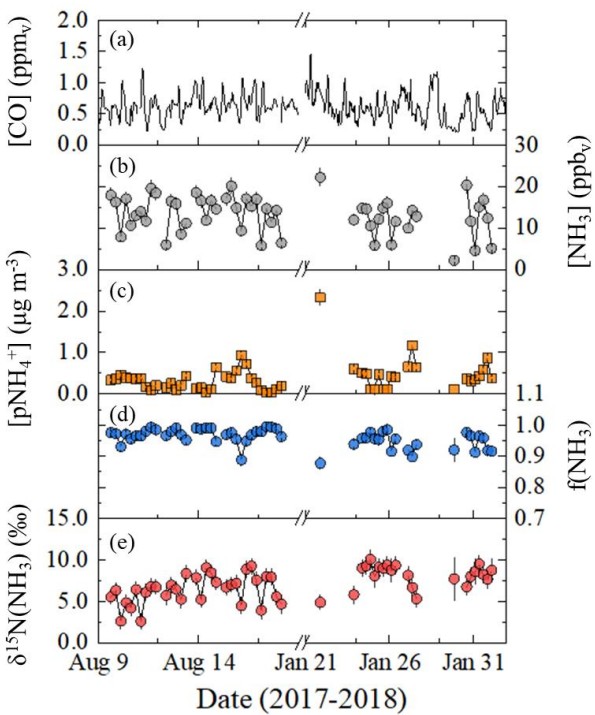

**Figure 1: Near-highway (Providence, RI, US) data summary of (a) [CO] (1 h average) (b) [NH$_3$], (c) [pNH$_4^+$], (d) f(NH$_3$) (=[NH$_3$](mol)/[NH$_x$](mol)), and (e) δ$^{15}$N(NH$_3$). The NH$_x$ data was generated from an active collection technique using a denuder-filter pack with a collection time of 6 h, and the error bars for concentrations and δ$^{15}$N(NH$_3$) measurements shown as black vertical lines represent the RSD (%) and propagated error, respectively. The break in the x-axis separates the summer and winter measurements.**





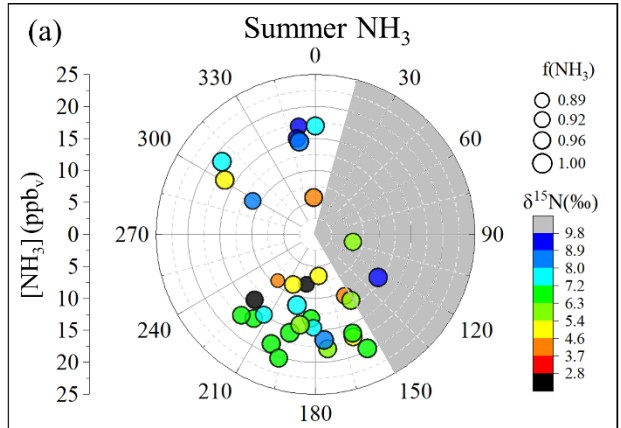
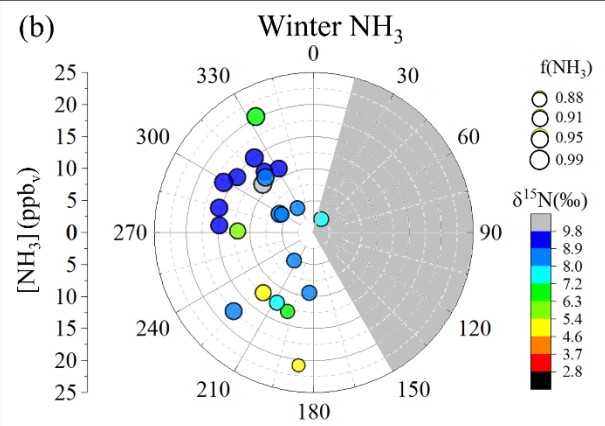

**Figure 2: Wind sector analysis of samples collected at the near-highway monitoring site (Providence, RI, US) for [NH₃] (circles) in (a) summer and (b) winter. The data is size-coded for f(NH₃) and color-coded for δ¹⁵N(NH₃) (‰). The monitoring location is downwind of I-95 except for wind directions 15 to 150° (grey-shaded region).**




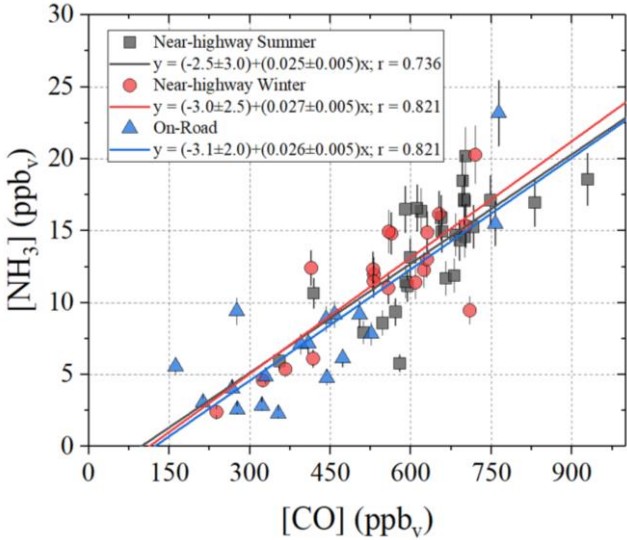

**Figure 3: Linear relations between [NH₃] and [CO] from the near-highway (Providence, RI, US) and mobile on-road (northeastern US) measurements. The [NH₃] data were based on acid-coated denuder collection and the [CO] represents the average of the on-line determined concentrations over the collection period. The linear regressions (solid lines) and Pearson's correlation coefficients (r) are provided for each respective measurement location.**


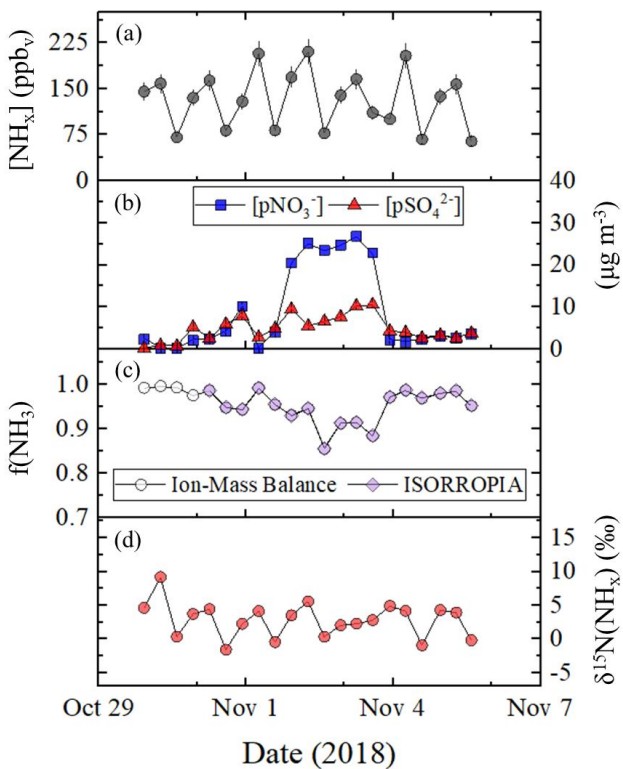

**Figure 4: Tunnel (Shenyang, Liaoning, China) data summary of (a) [NH$_x$], (b) concentrations of [pNO$_3$⁻] (blue square) and [pSO$_4$²⁻] (red triangle), (c) f(NH$_3$) calculated using ion-mass balance (open circle) and modelled using ISORROPIA (purple diamond), and (d) δ¹⁵N(NH$_x$). The data was generated from using a denuder-filter pack with a collection time of approximately 8 h. Error bars for concentrations and δ¹⁵N(NH$_x$) measurements shown as black vertical lines represent the RSD (%) and propagated error, respectively. ISORROPIA was not run for five collection periods, due to the absence of relative humidity and temperature data.**



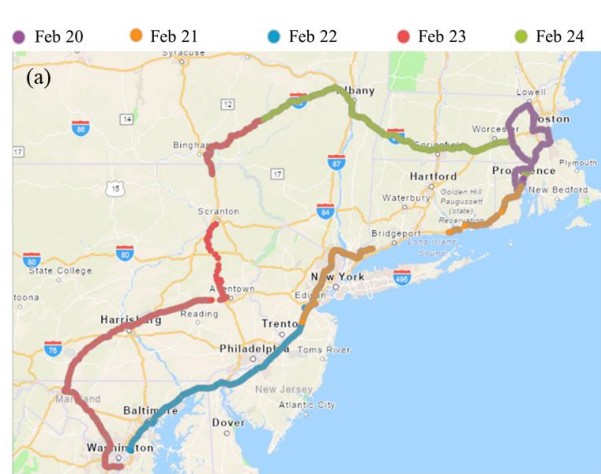

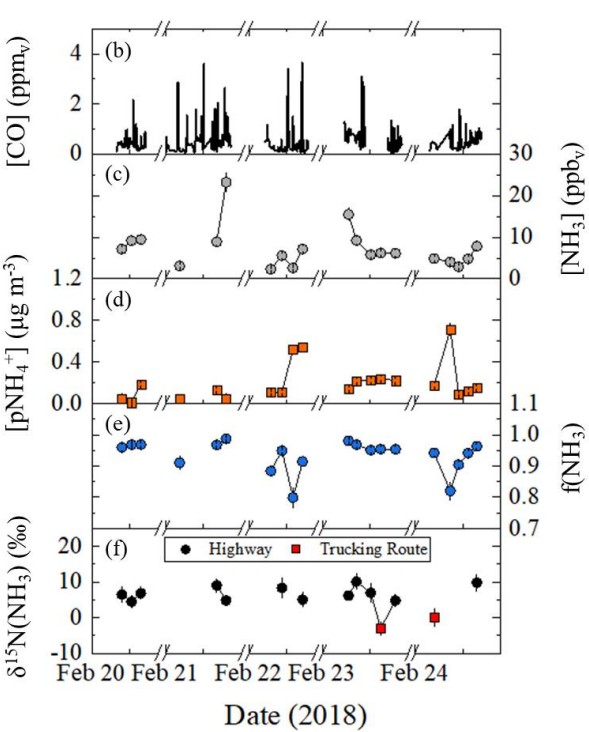


**Figure 5: Mobile on-road (northeastern US) measurements including (a) spatial mapping of measurement path sorted by date and data summary of (b) [CO] (1-min average) (c) [NH₃], (d) [pNH₄⁺], (e) f(NH₃) (=[NH₃](mol)/[NH$_x$](mol)), (f) $\delta^{15}$N(NH₃) for highway (black circle) and trucking routes (red square). The NH$_x$ data was generated using a denuder-filter pack with a collection time of approximately 1 h, and the error bars for concentrations and $\delta^{15}$N(NH₃) measurements shown as black vertical lines represent the**
**RSD (%) and propagated error, respectively. The break in the x-axis separates breaks in the mobile measurements. Image (a) was created using ArcGIS Copyright ⓒ 1995-2019 Esri.**



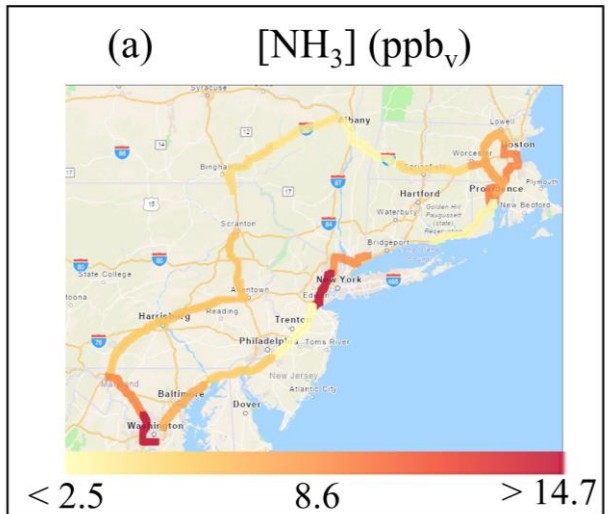
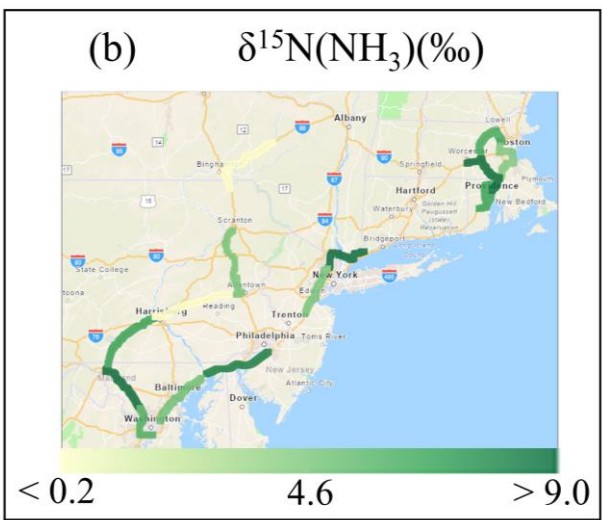

**Figure 6: Spatial maps of (a) mean [NH₃] (ppbᵥ) and (b) δ¹⁵N(NH₃) (‰) from on-road collections in the northeastern US. Each color represents one concentration or isotope measurement for NH₃ collected over a highway segment at an approximate 1 h resolution using an acid-coated denuder. Note that there are fewer reported δ¹⁵N(NH₃) values than [NH₃] because some samples had an elevated blank (i.e., f_Blank > 30 %) and were not measured for δ¹⁵N(NH₃). Images were created using ArcGIS Copyright ©1995-2019 Esri.**



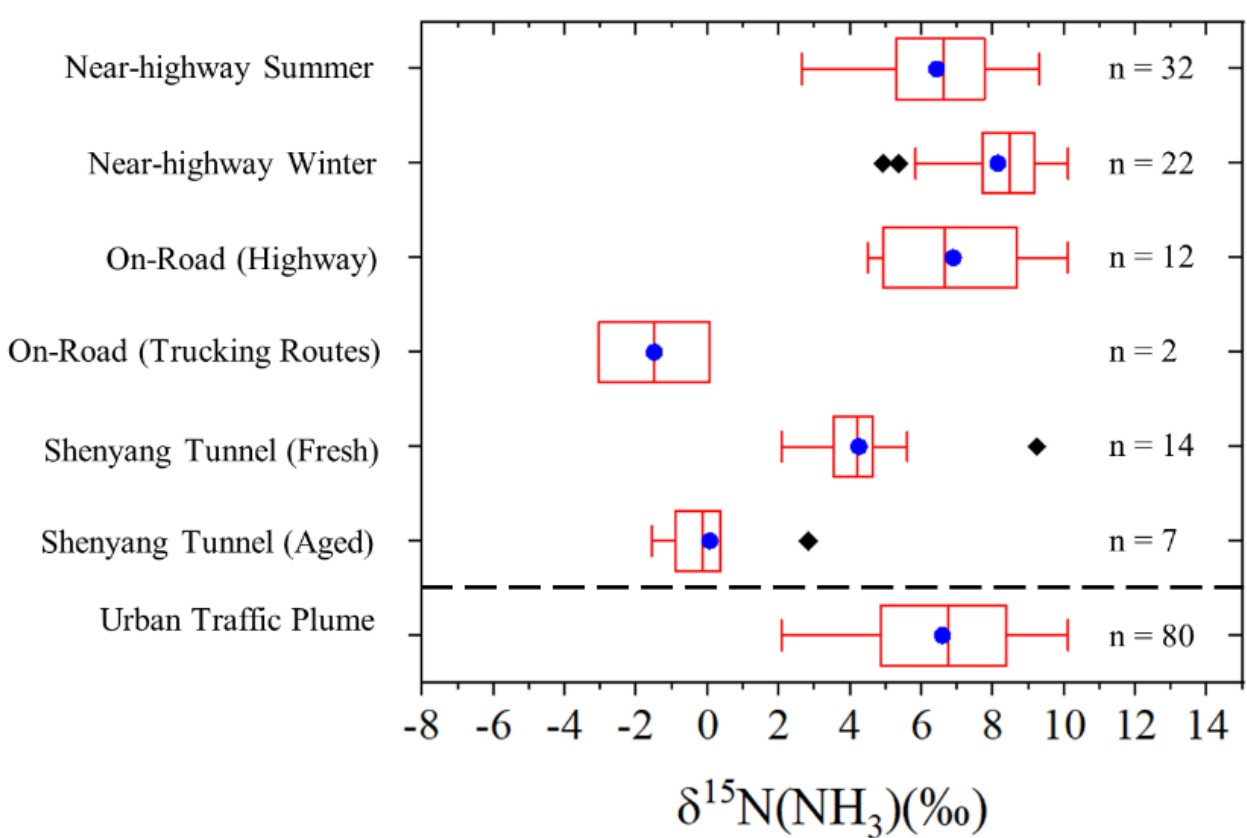

**Figure 7: Box and whisker plot summarizing the distribution (lower extreme, lower quartile, median (blue circle), upper quartile,**
**upper extreme, and outliers (black diamond)) of $\delta^{15}N(NH_3)$ measurements from near-highway, on-road, and tunnel sampling. The "Urban Traffic Fresh Plume" category represents the combination of $\delta^{15}N(NH_3)$ measurements from the near-highway, on-road (highway), and Shenyang tunnel (fresh) sampling.**
