# Peer review of "Characterizing the Spatiotemporal Nitrogen Stable Isotopic Composition of Ammonia in Vehicle Plumes"

_Atmospheric Chemistry and Physics, 2020_

## Referee Comment (RC1) · Anonymous Referee #1 · 6 Apr 2020

This paper by Walters et al. entitled "Constraining ammonia emissions in vehicle plumes utilizing nitrogen stable isotopes" describes several measurement campaigns that occurred either in Northeastern US or in China aiming at measuring NH3 and pNH4+ (= NHx) mixing ratios and isotopic composition. The authors used an active collection technique (filters + denuders) previously tested in laboratory to ensure the complete collection of NHx. They compare results from this technique from observations using passive collection, both from this study and from the literature. They also discuss the variables that could explain the observed spatio-temporal variability in d15N-NH3, and finally provide an updated range of d15N-NH3 from vehicle emissions. I find that overall the paper is well written, and the findings are definitely worth of publication, provided that the authors address the mostly minor comments that follow. I

especially appreciated the care taken data quality check, and the detailed field and lab operating procedures.

My main concern relates more to the structure/length of the article. I think that overall it is a very long paper to highlight the main result, which really is the range of d15N-NH3 from vehicle emissions. I think this could be a much shorter paper, which could emphasize more on the importance of characterizing NH3 isotopic composition from vehicle to be used in prospective NH3 source apportionment studies, which will likely become the norm in the close future, as has been the case over the past 20 years for HNO3. I realize that this comment is probably not very constructive, but I was thinking that the whole active/passive collection technique comparison, while very interesting and certainly useful, could be the subject of a separate article that the authors could refer to here. That alone would considerably lighten the results/discussion, and would help the reader to follow more easily the different campaign results and subsequent discussion.

Detailed Comments:

I think the title is somewhat misleading. You don't really constrain the vehicle ammonia emissions using N isotopes. The title as is suggests a source apportionment study, which is not the case. It should read: "Characterizing the isotopic composition of ammonia from vehicle plumes" or something like that.

Your abstract makes no mention of the comparison between active and passive collection techniques, which supports my previous point that you could remove that from your manuscript and have it in a separate paper. It reads as a sideways discussion in the present format, and distracts the reader from the main findings. I am not saying it is not interesting and useful, just that it could be its own paper.

L. 37-39: I thought soil acidification is mostly due to HNO3. How can an alkaline compound like NH3 cause acidification?

L. 47: Helpful if you could indicate here NH3 atmospheric lifetime.

L. 48-49: The Templer group in Boston has more recent studies highlighting large vehicle contribution to urban NH3 budget. Check out the Decina et al. papers, particularly relevant since you drove to Boston for this study.

L. 61: Can you quantify here the contribution as a % at the global scale? L. 65: Once again, check the work lead by the Templer group in Boston about N deposition in urban areas.

L. 66-67: how are "fuel-combustion" and "vehicle" sources different? Isn't the latest included within the first?

L. 90-91: Didn't you just say that these techniques were shown to not accurately capture the d15n-NH3, based on work by Skinner et al.? This seems contradictory.

L. 158: How long is the inlet line?

L. 209: Did you characterize potential inlet loss, and induced fractionation on NH3, to see if tit was indeed negligible?

L. 214: Any chance the denuders could trap a portion of the particulate phase as well, on top of the gas phase?

L. 214: Can you give quantify your detection limits?

L. 219: pNO3-, but what about pNH4+ ?

L. 241: What do you use the ethanol for?

L. 380: Does it mean that the urban background NH3 has the isotopic composition of vehicle emissions?

L. 395-401: I understand that you can't estimate f(NH3) accurately, but why can't you calculate the concentration of pNH4+ here? Were the Nylon filters also saturated? There is no mention of that aspect it, and it should be expanded on.

L. 409: An introduction sentence about what ISORROPIA is would be nice.

L. 421: I think it would be useful and interesting to provide, maybe in the SI, the isotopic composition for each component, especially the nylon-collected $pNH_4+$. And maybe expand on the different isotopic composition of NHx and $pNH_4+$, if such is the case (and I expect it to be).

L. 431: Section title should be revised; it is the same ast the previous section title

L. 481: Please recall here what are elevated NH3 concentrations.

L. 521-523: Maybe recall that your f(NO3) is approximate in this case.

––––––––––––––––––––––––––––––––

---

## Referee Comment (RC2) · Anonymous Referee #2 · 18 Jun 2020

This manuscript reports integrated offline measurements of gas phase ammonia and particle phase ammonium concentrations and d15N isotopic signatures, targeting airmasses that are strongly impacted by vehicle emissions. The combination of roadside and on-road data from the Eastern US, and tunnel data from China, provide confidence that the results are representative. The description of the sample collection and analysis procedures is very thorough. Because the gas phase fraction of the majority of the samples is very high, and the enhancements over background concentrations are high, the measured isotopic signatures can be interpreted as reflecting the emissions themselves. This allows the authors to report a narrow range of isotopic signature for a source of ammonia (traffic) that is important, but poorly constrained, in urban areas. One of the most important and unambiguous results from the paper is the identifica-

tion of an offset in the isotopic signatures from the samples acquired through passive sampling. The authors provide clear theoretical support for the magnitude of this effect based on the relative diffusion rates for the isotopologues in air. Given the importance of this result, it should be mentioned in the abstract.

The topic and results are very appropriate for publication in ACP. Overall, the paper is well-written, though longer than necessary. For example, the section ruling out contributions from background NH3 (lines 569 – 585) does not need to be so long. I recommend that it be published after some minor revisions.

Specific comments

Line 71 – clarify whether improvements refers to the sources or our understanding of them

Line 76-78 This sentence is worded unclearly

Line 268 -269 Is this sentence saying that the limit of detection for this method was higher than usual due to contamination? It's hard to follow the logic.

Line 319 Define what is meant by fblank

Line 430 Section 3.1.3 appears to have the wrong title

Figure 7 – showing a median and interquartile range for two samples seems a bit excessive. Perhaps just report the two values as individual symbols.

Line 671 – Smirnoff is misspelled

---

## Author Comment (AC1) · 22 Jul 2020

We appreciate the helpful comments and feedback from Reviewer #1, which have helped improve our manuscript. We have carefully considered the recommendations of Reviewer #1 to shorten the manuscript, which was also suggested by Reviewer #2. To this end, we have shortened our introduction, moved the description of our denuder and filter preparation, handling, and extraction protocol to the supplement, and removed our discussion of the elevated vehicle [$NH_3$], which distracted from our main point of characterizing the isotopic composition of vehicle derived $NH_3$. Overall, these changes have shortened the manuscript by ~150 lines.

Reviewer #1 also pointed out that perhaps we should consider reporting the results of passive vs active collection for $\delta^{15}N(NH_3)$ characterization in a separate manuscript. We respectfully disagree with this suggestion and feel that this is an important result for this study, as pointed out by Reviewer #2. Specifically, this comparison helps put our measurements into context with previous studies that have reported very different $\delta^{15}N(NH_3)$ values derived from vehicle emissions. Our observation that a large $\delta^{15}N(NH_3)$ offset exists between active and passive sampling techniques reconciles differences in our measurements with previous literature reports and highlights the need for the reactive nitrogen isotope community to consider using robust, laboratory and field verified techniques shown to be accurate in characterizing $\delta^{15}N$. This is an incredibly important point that cannot be stressed enough. Therefore, we did not remove our comparison between active and passive $NH_3$ collection in the revised manuscript. Below we provide a point-by-point response to specific comments raised by Reviewer #1.

Specific Comments:

**Comment:** I think the title is somewhat misleading. You don't really constrain the vehicle ammonia emissions using N isotopes. The title as is suggests a source apportionment study, which is not the case. It should read: "Characterizing the isotopic composition of ammonia from vehicle plumes" or something like that.

**Response:** Thank you for this comment. We have changed the title in the revised manuscript to "Characterizing the Spatiotemporal Nitrogen Stable Isotopic Composition of Ammonia in Vehicle Plumes" to reflect the content of this work better.

**Comment:** Your abstract makes no mention of the comparison between active and passive collection techniques, which supports my previous point that you could remove that from your manuscript and have it in a separate paper. It reads as a sideways discussion in the present format and distracts the reader from the main findings. I am not saying it is not interesting and useful, just that it could be its own paper.

**Response:** We appreciate the comment, but we believe this comparison is an important finding and should be included in this manuscript, which is also aligned with the opinion of Reviewer #2. Thank you for pointing out that we did not mention this comparison in our abstract, which was an oversight, and also brought up by Reviewer #2. In the revised manuscript, we have changed the abstract to draw attention to the comparison between active and passive sampling, adding the following sentences, "Our recommended vehicle $\delta^{15}N(NH_3)$ signature is significantly different

from previous reports. This is due to a large and consistent $\delta^{15}N(NH_3)$ bias of approximately -15.5 ‰ between commonly employed passive $NH_3$ collection techniques and the laboratory-tested active $NH_3$ collection technique." Additionally, we have restructured our section headings to help reduce the comparison between active and passive sampling to read as a sideways discussion. We have removed the subsection division in the results and discussion section between (1) Active $NH_x$ Collection using a Denuder-Filter Pack and (2) Comparison Between Active and Passive Collection. We feel that this reduces the complicated subsection grouping and creates more streamlined results and discussion sections in the revised manuscript.

**Comment:** L37-39: I thought soil acidification is mostly due to HNO3. How can an alkaline compound like NH3 cause acidification?

**Response:** There are numerous processes in which $NH_3$ can cause soil acidification that has been well-documented, including plant uptake and assimilation, nitrification, and $NH_3$ volatilization. The uptake and assimilation of $NH_4^+$ results in a net release of $H^+$ as $NH_4^+$ is deprotonated during this process. Nitrification associated with the oxidation of $NH_4^+$ to $NO_2^-$ and subsequent oxidation to $NO_3^-$ will also lead to the net release of $H^+$:

$$NH_4^+ + 2O_2 \rightarrow NO_3^- + H_2O + 2H^+$$

Finally, during $NH_3$ volatilization, the pH of the soil surface will decrease as $H^+$ is released when $NH_4^+$ is converted to $NH_3$:

$$NH_4^+ \leftrightarrow NH_3 + H^+$$

To clarify how $NH_{3/4}$ can lead to acidification we have revised the sentence in question to the following, "Deposition of $NH_3$ and its secondary product, particulate ammonium ($pNH_4^+$), have critical environmental consequences, including soil acidification (via plant assimilation, nitrification, and $NH_3$ volatilization), eutrophication, and decreased biodiversity in sensitive ecosystems," and included an additional reference that does an excellent job reviewing soil acidification (Bolan, N.S., Hedley, M.J., White, R.E. Processes of soil acidification during nitrogen cycling with emphasis on legume based pastures. Plant and Soil, 134(1), 53-63, 1991.

**Comment**: L47: Helpful if you could indicate here NH3 atmospheric lifetime.

**Response:** Thank you for pointing this out. We have revised this sentence to include the $NH_3$ atmospheric lifetime to the following, "While agricultural activities are known to dominate the emission of $NH_3$, accounting for over 60 % of the global inventory (Bouwman et al., 1997), there are significant spatiotemporal variabilities due to its short atmospheric lifetime that is on the order of a several hours to a day (Paulot et al., 2016) and its multitude of emission sources (e.g., Hu et al., 2014)."

**Comment**: L48-49: The Templer group in Boston has more recent studies highlighting large vehicle contributions to urban NH3 budget. Check out the Decina et al., papers, particularly relevant since you drove to Boston for this study

**Response**: We have added the following reference to the end of this sentence in the revised manuscript, "Decina, S. M., Templer, P. H., Hutyra, L. R., Gately, C. K. and Rao, P.: Variability, drivers, and effects of atmospheric nitrogen inputs across an urban area: emerging patterns among human activities, the atmosphere, and soils, Sci. Total Environ., 609, 1524–1534, 2017."

**Comment**: L61: Can you quantify here the contribution as a % at the global scale?

**Response**: To help shorten the manuscript, we have removed this sentence in the revised manuscript. We point out that vehicle emissions are an important urban source of $NH_3$, "In urban regions, vehicle derived emissions have been identified as a major $NH_3$ source (Decina et al., 2017; Gong et al., 2011; Li et al., 2006; Livingston et al., 2009; Meng et al., 2011; Nowak et al., 2012; Sun et al., 2014, 2017). Recently, vehicle $NH_3$ emissions have been suggested to be a key driver of N deposition in urban and urban-affected regions (Fenn et al., 2018)."

**Comment**: L65: Once again, check the work lead by the Templer group in Boston about N deposition in urban areas

**Response**: Thank you for this comment. We have included additional references to Decina et al., 2017 and Decina, S. M., Hutyra, L. R. and Templer, P. H.: Hotspots of nitrogen deposition in the world's urban areas: a global data synthesis, Front. Ecol. Environ., 18(2), 92–100, 2020.

**Comment**: L66-67: how are "fuel-combustion" and "vehicle" source different? Isn't the latest included with the first?

**Response**: The original use of "fuel-combustion" was to refer to stationary fuel-combustion, such as electricity and heating generation. To improve clarity, we have changed "fuel-combustion" to "stationary fuel-combustion" in the revised manuscript.

**Comment**: L90-91 Didn't you just say that these techniques were shown to not accurately capture the δ15N-NH3, based on work by Skinner et al.,? This seems contradictory.

**Response**: Thanks for the comment, and we have removed this sentence in the revised manuscript.

**Comment**: L. 158: How long is the inlet line?

**Response**:  No inlet sampling line was used for the employed denuder-filter pack (ChemComb Speciation Cartridge), as the cartridges were directly exposed to ambient air as stated in section 2.2, " The samplers were directly exposed to ambient air without the use of an additional inlet tubing to prevent the loss of NH$_3$." The inlet described in Line 158 of the original manuscript refers to the air inlet of the ChemComb Speciation Cartridge, where ambient air is first introduced into the sampler.  This piece is approximately 4 cm long, which has been indicated in the revised manuscript in Seciton 2.2, "The PTFE coated air inlet (~4 cm)…"

**Comment**:  L.209:  Did you characterize the potential inlet loss, and induced fractionation on NH3, to see if it was indeed negligible?

**Response**:  Thank you for pointing this out, as we also feel that it is very important to consider potential inlet losses of reactive species to accurately characterize isotopic compositions, which had been previously largely ignored.  In a previous study, we have conducted extensive laboratory experiments to document such potential sampling artifacts (Walters, W.W., Hastings, M.G. Collection of ammonia for high time-resolved nitrogen isotopic characterization utilizing an acid-coated honeycomb denuder.  Anal. Chem., 90, 8051-8057, 2018).  We evaluated the potential of the ChemComb inlet to induce fractionation by comparing NH$_3$ collections with (1) honeycomb denuders housed in the ChemComb Cartridge and (2) a gas scrubbing impinger that does not have an inlet (control), in which the NH$_3$ line was directly scrubbed in an acid solution.  We found no statistical difference in $\delta^{15}$N(NH$_3$) between the ChemComb sampler and the control, suggesting that inlet loss as a potential source of $\delta^{15}$N(NH$_3$) fractionation was negligible.  This result is also in agreement with Koutrakis et al., 1993 that reported no evidence for significant loss of NH$_3$ induced via the PTFE-coated sampling inlet.  We have added the following sentence to the revised manuscript in Section 2.2 to demonstrate that we have considered the potential influence of the sampling inlet on inducing $\delta^{15}$N(NH$_3$) fractionation, "The PTFE coated inlet has been shown to lead to a negligible loss of NH$_3$ and induce insignificant $\delta^{15}$N(NH$_3$) fractionation (Koutrakis et al., 1993; Walters and Hastings, 2018)."

**Comment**:  L.214:  Any chance the denuders could trap a portion of the particulate phase as well on top of the gas phase?

**Response**:  Thank you for raising this point.  Particulates do not contribute to the final measurement from the denuder extracts due to the system design of the denuder-filter pack and operation conditions.  We have added the following to the revised manuscript in section 2.2 to elaborate on this point, "Briefly, ambient air is drawn into the sampler and reactive gases are removed under laminar flow conditions such that radial mixing can only be achieved via diffusion-based processes.  Particulates, with their much lower diffusion velocity compared to gases, cannot migrate to the walls of the denuder during the residence time within the unit and are collected on a downstream filter pack.  The samplers are also held vertically to limit the potential for gravitational settling of particles onto the denuder surfaces, such that particulates do not contribute to the denuder extract (Ali et al., 1989)."

**Comment**: L214: Can you give quantify your detection limits?

**Response**: Thank you for this comment. Limits of detection are based on off-line ion quantification as described in Section 2.3 in the revised manuscript. We have added the limit of detection (LOD) quantification for $NH_4^+$, $NO_2^-$, $NO_3^-$, and $SO_4^{2-}$ in section 2.3, "The limit of detection (LOD) of the quantified ions were no higher than 0.5, 0.2, 2.0, and 1.5 $\mu mol \cdot L^{-1}$ for $[NH_4^+]$, $[NO_2^-]$, $[NO_3^-]$, and $[SO_4^{2-}]$, respectively". To improve the clarity of the manuscript, we have moved the sentence on Line 214 in the original manuscript to after the instrumentation LOD was discussed in the revised manuscript.

**Comment**: L.219: pNO3-, but what about pNH4+?

**Response**: Thank you for raising this point. Nylon filters will quantitatively capture $pNO_3^-$ but a significant fraction of $pNH_4^+$ will volatilize off this type of filter (see Walters, W.W, Blum, D.E., Hastings, M.G. Selective collection of particulate ammonium for nitrogen isotopic characterization using a denuder-filter pack sampling system, Anal. Chem, 91, 7586-7594, 2019 & Yu, X., Lee, T., Ayres, B., Kreidenweis, S.M., Malm, W., Collett, J.L. Loss of fine particulate ammonium from denuded nylon filters. Atmos Environ, 40, 4797-4807, 2006). However, a backup acid-coated filter will quantitatively capture any volatilized $pNH_4^+$ (Walters et al., 2019). We note that we had originally planned to quantify the inorganic anions collected on the filters in all measurement campaigns, which is why we planned to utilize both Nylon and citric acid coated filters. However, we found the extracted anion concentrations to be below our detection limits, such that this data was not reported for the stationary and mobile US measurements. If quantification of $pNH_4^+$ is the main goal (for concentration or isotopic analysis), a single acid-coated filter downstream from an acid-coated denuder should suffice, as we have pointed out in Walters et al., 2019.

We clarified this sentence in the revised manuscript to the following, "However, due to potential loss of semi-volatile $NH_4NO_3$, all subsequent campaigns utilized a Nylon filter (Cole-Parmer, 0.8 $\mu m$ pore, 47 mm diameter) which has been shown to collect and retain $pNO_3^-$ quantitatively (Yu et al., 2005). A significant fraction of $pNH_4^+$ collected on denuded Nylon filters may volatilize (Yu et al., 2006), such that a backup acid-coated (5 % citric acid (w/v) in water) cellulose filter (Whatman, 8 $\mu m$ pore, 47 mm diameter) is used to capture any volatilized $NH_3$ from the collected particles and/or $NH_3$ breakthrough during conditions of denuder saturation (Walters et al., 2019)."

**Comment**: L.241: What do you use ethanol for?

**Response**: This was used to wet the hydrophobic Teflon filter surface. We have clarified this sentence in the revised manuscript, "The PTFE filters were pre-wetted with 500 $\mu L$ of ethanol to wet its hydrophobic surface before extraction." We note that this text was moved to the Supplement (Text S1) in the revised manuscript to shorten the manuscript length.

**Comment**: L.380: Does it mean that the urban background NH3 has the isotopic composition of vehicle emissions?

**Response**: Thank you for this comment. I think concluding that urban background $\delta^{15}N(NH_3)$ is the same as vehicle emissions based on wind direction analysis at the near-highway stationary site would be incorrect since the measurement location is near a major $NH_3$ emission source. We have pointed this out in the revised manuscript, "Overall, the $\delta^{15}N(NH_3)$ values were not found to be significantly different when the monitoring site was upwind or downwind of I-95, with averages of 7.6±1.4 ‰ (n=3) and 7.1±1.8 ‰ (n=51), respectively (p>0.05), which is likely due to the proximity of the sampling location to airmasses significantly influenced by vehicle emissions."

**Comment**: L.395-401: I understand that you can't estimate f(NH3) accurately, but why can't you calculate the concentration of pNH4+ here? Were the Nylon filters also saturated? There is not mention of that aspecit it, and it should be expanded on.

**Response**: The Nylon filters were likely not "saturated", but $pNH_4^+$ collected on Nylon filters are subject to significant volatilization, as we have mentioned in section 2.2. Thus, we cannot quantitatively determine the $NH_x$ speciation as $pNH_4^+$ extracted from the Nylon filter likely contains a negative artifact. The extracted $pNH_4^+$ extracted from the acid-coated filter represents both $NH_3$ breakthrough due to denuder saturation as well as some component of $pNH_4^+$ volatilization. We have further clarified why we can't quantitatively determine $pNH_4^+$ in section 3.2 of the revised manuscript, "Thus, our $NH_x$ measurements are expected to be accurate, but there could be uncertainty in the $NH_x$ speciation, because the $NH_4^+$ extracted from the acid-coated denuder and Nylon filter will have a low bias due to denuder saturation and $pNH_4^+$ volatilization, respectively, and $NH_4^+$ extracted from the acid-coated filter will derive from both $NH_3$ breakthrough and $NH_3$ volatilized from the Nylon filter."

**Comment**: L.409: An introduction sentence about what ISORROPIA is would be nice.

**Response**: In the revised manuscript, we have further elaborated on ISORROPIA as followed, "$NH_x$ speciation was also estimated using ISORROPIA, which is a gas-aerosol equilibrium partitioning model (Fountoukis and Nenes, 2007; Nenes et al., 1998)."

**Comment**: L.421: I think it would be useful and interesting to provide, maybe in the SI, the isotopic composition for each component, especially the nylon-collected pNH4+. And maybe expand on the different isotopic compositions of NHx and pNH4+, if such is the case (and I expect it to be).

**Response**: Thank you for this comment as this is an interesting point and one of the original goals of attempting to collect and speciate between $NH_3$ and $NH_4^+$ simultaneously. However, due to our

$NH_x$ speciation problems in the tunnel measurements from $NH_3$ denuder saturation (as we have well-documented in section 3.2), it is impossible to relate the $NH_4^+$ extracted from the acid-coated denuder, Nylon filter, and acid-coated filter to their atmospheric component due to numerous sampling artifacts. Therefore, we do not think it would be a good idea to discuss the $\delta^{15}N(NH_4^+)$ from the varying sampling media and attempt to relate them to $NH_3$ and $pNH_4^+$, and this was the reason we presented the results in section 3.2 as $\delta^{15}N(NH_x)$. As requested, we have included a figure in the Supplement (Fig. S6) that shows the varying $\delta^{15}N$ values of $NH_4^+$ extracted from the acid-coated denuder, Nylon filter, and acid-coated filter in the revised manuscript. In section 3.2 of the revised manuscript, we added the average $\delta^{15}N$ values of the varying sampling media, "The measured $\delta^{15}N$ from $NH_4^+$ extracted from the acid-coated denuders, Nylon filters, and acid-coated filters averaged 6.0±5.6 ‰ (n=21), 1.0±10.7 ‰ (n=21), and -20.0±10.1 ‰ (n=21) (Figure S6)." Additionally, we note, "These $\delta^{15}N$ differences to some degree reflect differences in the $\delta^{15}N$ of ambient $NH_3$ and $NH_4^+$ but are difficult to interpret due to the ambiguity in $NH_x$ speciation." Since we have strong evidence that we captured 100% of $NH_x$, but did not accurately speciate $NH_x$, we focus our discussion in the text on $\delta^{15}N(NH_x)$.

**Comment**: L.431: Section title should be revised; it is the same as the previous section title

**Response**: Thank you for pointing this out. We have provided the correct subtitle name, "Mobile On-Road $NH_3$ Survey in Northeastern US" in the revised manuscript.

**Comment**: L.481: Please recall here what are elevated NH3 concentrations.

**Response**: Thank you for this comment. In the revised manuscript, we removed our discussion of the elevated vehicle [$NH_3$] to reduce the manuscript length and to draw attention to our $\delta^{15}N$ results, which is the main focus of this work.

**Comment**: L.521-523: Maybe recall that your f(NO3) is approximate in this case.

**Response**: Thank you for pointing this out. We believe Reviewer #1 is referring to f($NH_3$) and not f($NO_3$), and have included that the f($NH_3$) in our tunnel measurements were an approximation in the revised manuscript, "The temporal tunnel variability is not likely to be driven by f($NH_3$) partitioning influences as the estimated f($NH_3$) was not found to be significantly different between periods the tunnel was open or closed (p>0.05), indicating a significant change in $NH_3$/$pNH_4^+$ partitioning did not occur during these periods."

---

## Author Comment (AC2) · 22 Jul 2020

Thank you for your constructive feedback, comments, and suggestions, which have helped improve our manuscript. We agree that our finding of a large $\delta^{15}N$ offset between active and passive collection is significant, and we have revised our abstract to draw attention to this finding. In particular, we have added the following sentences, "Our recommended vehicle $\delta^{15}N(NH_3)$ signature is significantly different from previous reports. This is due to a large and consistent $\delta^{15}N(NH_3)$ bias of approximately -15.5 ‰ between commonly employed passive $NH_3$ collection techniques and the laboratory-tested active $NH_3$ collection technique," and added, "This work… and highlights the importance of utilizing verified collection methods for accurately characterizing $\delta^{15}N(NH_3)$ values," to the abstract.

We understand the concerns about the length of the manuscript, which was also raised by Reviewer #1. As suggested, we have simplified our discussion of background $NH_3$ influences from our on-road measurements in the revised manuscript to the following, "Furthermore, we do not expect background $NH_3$ contributions to have played a significant role in the spatial $\delta^{15}N(NH_3)$ variability observed from the on-road measurements in the Northeastern US. While lower $\delta^{15}N(NH_3)$ values in non-urban regions might be consistent with an increased contribution from background agricultural emissions which tend to have a low $\delta^{15}N(NH_3)$ signature (e.g., -31 to -14 ‰; Hristov et al., 2011), we expect these temperature-dependent emissions to be minimal during the winter when the on-road measurements were conducted." Additionally, we have shortened our introduction, moved the description of our denuder and filter preparation, handling, and extraction protocol to the supplement, and removed our discussion of the elevated vehicle [$NH_3$], which distracted from our main point of characterizing the isotopic composition of vehicle derived $NH_3$. Overall, these changes have shortened the manuscript by ~150 Lines. Below we provide a point-by-point response to specific comments raised by Reviewer #2:

**Comment:** Line 71 – clarify whether improvements refers to the sources or our understanding of them

**Response:** Here we aim to make the point that while $\delta^{15}N(NH_3)$ might be a potentially valuable tool for tracking $NH_3$ emissions, the number of $\delta^{15}N(NH_3)$ source characterization studies are limited. Thus, we need to enhance our $\delta^{15}N(NH_3)$ emission inventory before we can begin to utilize $\delta^{15}N(NH_3)$ as a quantitative tool for source apportionment. To clarify this point, we have changed this sentence in the revised manuscript to, "However, $\delta^{15}N(NH_3)$ source characterization studies are limited, particularly for non-agriculture $NH_3$ emissions (Chang et al., 2016; Felix et al., 2013; Freyer, 1978; Heaton, 1987; Smirnoff et al., 2012); thus, to quantitatively utilize this tracer for $NH_3$ source apportionment requires further improvements in $\delta^{15}N(NH_3)$ source emission signatures and an increased understanding of spatiotemporal variabilities."

**Comment**: Line 76-78 – This sentence is worded unclearly.

**Response:** Thank you for pointing this out. We have simplified this sentence in the revised manuscript as follows, "To quantitatively utilize $\delta^{15}N(NH_3)$ for $NH_3$ source apportionment requires distinguishable isotopic signatures, such that we need to understand the drivers behind the reported large variability in $\delta^{15}N(NH_3)$ from vehicle emissions."

**Comment:** Line 268-269 Is this sentence saying that the limit of detection for this method was higher than usual due to contamination? It's hard to follow the logic.

**Response:** When using the BrO-/azide chemical method for converting $NH_4^+$ to $N_2O$, we find a significant reagent $N_2O$-blank. This reagent blank makes it difficult to accurately and precisely characterize $\delta^{15}N$ for low concentration samples, such that we only conducted $\delta^{15}N$ analysis for samples with an $[NH_4^+] > 2$ µmol·L$^{-1}$. We have clarified this point in the revised manuscript to the following, "Briefly, $\delta^{15}N(NH_4^+)$ was measured based on an established off-line wet-chemistry technique involving hypobromite (BrO$^-$) oxidation and acetic acid/sodium azide reduction (Zhang et al., 2007), which was conducted for samples with $[NH_4^+] > 2$ µmol·L$^{-1}$."

**Comment:** Line 319 Define what is meant by fblank

**Response:** The $f_{Blank}$ refers to the fraction of collected $NH_4^+$ that corresponds to the field blank. We have defined $f_{Blank}$ in the text and rewrote the sentence in the revised manuscript to the following, "Blank $\delta^{15}N(NH_4^+)$ corrections were made for all samples when the fraction of the field blank ($f_{Blank} = [NH_4^+]_{blank}/[NH_4^+]_{total}$) were less than 30% of the total collected $NH_4^+$, as the propagated $\delta^{15}N$ uncertainty generally did not exceed ±2.5 ‰ for this $f_{Blank}$ value."

**Comment:** Line 430 Section 3.1.3 appears to have the wrong title

**Response:** Thank you for pointing this out. The correct subtitle "Mobile On-Road $NH_3$ Survey in Northeastern US" and has been changed appropriately in the revised manuscript.

**Comment:** Figure 7 – showing a median and interquartile range for two samples seems a bit excessive. Perhaps just report the two values as individual symbols.

**Response:** We agree and have adjusted Figure 7 in the revised manuscript, such that "On-Road (Trucking Routes)" only shows the two data points and not a statistical summary.

**Comment**: Line 671-Smirnoff is misspelled

**Response:** Thank you for pointing this out. We have fixed this mistake in the revised manuscript.